# Segment then Splat: Unified 3D Open-Vocabulary Segmentation via Gaussian Splatting

**Yiren Lu**[1], **Yunlai Zhou**[1], **Yiran Qiao**[1], **Chaoda Song**[1], **Tuo Liang**[1],
**Jing Ma**[1], **Huan Wang**[2], **Yu Yin**[1][✉]
[1]Case Western Reserve University
[2]Westlake University
[1]{yiren.lu, yunlai.zhou, yiran.qiao, chaoda.song,
tuo.liang, jing.ma5, yu.yin}@case.edu
[2]wanghuan@westlake.edu.cn
https://vulab-ai.github.io/Segment-then-Splat/

## Abstract

Open-vocabulary querying in 3D space is crucial for enabling more intelligent perception in applications such as robotics, autonomous systems, and augmented reality. However, most existing methods rely on 2D pixel-level parsing, leading to multi-view inconsistencies and poor 3D object retrieval. Moreover, they are limited to static scenes and struggle with dynamic scenes due to the complexities of motion modeling. In this paper, we propose *Segment then Splat*, a 3D-aware open vocabulary segmentation approach for both static and dynamic scenes based on Gaussian Splatting. *Segment then Splat* reverses the long-established approach of "segmentation after reconstruction" by dividing Gaussians into distinct object sets before reconstruction. Once reconstruction is complete, the scene is naturally segmented into individual objects, achieving true 3D segmentation. This design eliminates both geometric and semantic ambiguities, as well as Gaussian–object misalignment issues in dynamic scenes. It also accelerates the optimization process, as it eliminates the need for learning a separate language field. After optimization, a CLIP embedding is assigned to each object to enable open-vocabulary querying. Extensive experiments on various datasets demonstrate the effectiveness of our proposed method in both static and dynamic scenarios.

## 1 Introduction

3D open-vocabulary querying marks a pivotal step in language-driven interaction with 3D environments, removing the need for predefined labels. This capability is vital for large-scale scene exploration, scene understanding, robotic navigation [1–3] and manipulation [4–7], where free-form text bridges human language and machine perception.

3D Gaussian Splatting (3DGS) [8] has been a widely adopted 3D representation due to its efficient training and real-time rendering capabilities. While 3DGS has demonstrated remarkable performance in scene reconstruction and novel view synthesis, it lacks inherent semantic understanding, limiting its applicability in tasks that require natural language-driven retrieval and reasoning.

To solve this issue, most existing works [9–12] incorporate a separate language field alongside Gaussian Splatting reconstruction. By rendering the language field into 2D feature maps, they enable pixel-based querying by retrieving relevant pixels based on the input text embedding. However, this approach essentially performs segmentation in 2D space rather than partitioning Gaussians in 3D space, leading to several drawbacks: 1) **Inconsistent 2D segmentation** across different views, leading

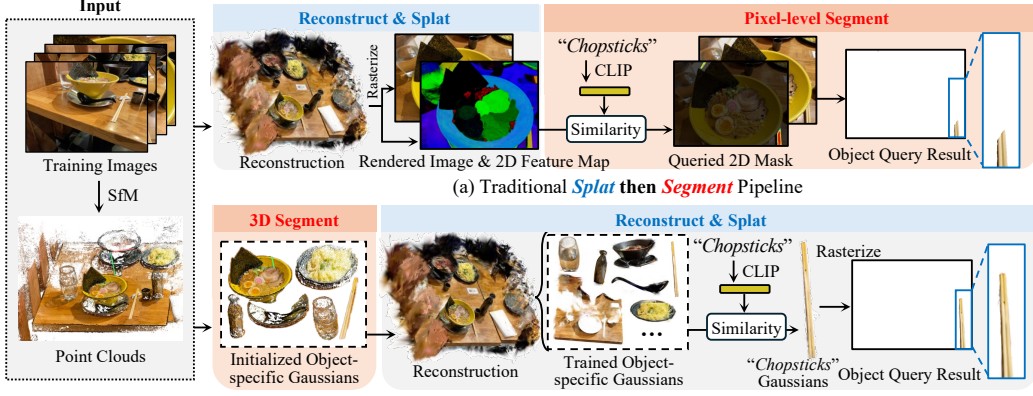

Figure 1: **Traditional 3D Open-Vocabulary Segmentation vs. our Segment-then-Splat Pipeline.**
(a) The traditional Splat-then-Segment pipeline learns a language field alongside the reconstruction of the entire scene. During object queries, it renders Gaussian language embeddings into a 2D feature map to identify relevant pixels based on the input text embedding. (b) In contrast, our Segment-then-Splat pipeline first initializes Gaussians into object-specific sets before reconstruction, ensuring a more precise object-Gaussian correspondence and improving segmentation accuracy.

to inaccurate object boundaries. 2) **Failure to capture true 3D object information**, complicating 3D object extraction and limiting downstream tasks like robot navigation and 3D manipulation. 3) **Inapplicability to dynamic scenes**, since Gaussians may have varying semantic meanings at different time steps, preventing straightforward extensions to time-varying or moving objects.

Recently, a few works [13, 14] have explored direct segmentation in 3D space. However, these approaches require a predefined number of objects for clustering [13] or are limited to foreground segmentation [14], and also cannot be directly applied to dynamic scenes.

Despite the differences in segmentation strategies, whether pixel-based or 3D-based, all existing approaches follow a "reconstruction then segmentation" (i.e., splat then segment) paradigm. This approach inherently results in imprecise object boundaries, as each Gaussian may encode geometric and semantic information from multiple objects, leading to geometric and semantic ambiguity.

In this paper, we propose *Segment then Splat*, a unified framework for 3D-aware open-vocabulary segmentation that can be applied to both static and dynamic scenes. Unlike existing methods that adopt a "splat then segment" approach, our method reverses the process by first initializing each object with a specific set of Gaussians, as shown in Fig. 1. During training, each set of Gaussians is assigned a unique object ID and contributes only to its corresponding object, guided by 2D multi-view mask supervision. By doing so, each Gaussian is dedicated to a single object and thus learns more accurate object geometry. Moreover, since the Gaussian-object correspondence is strictly maintained, our method can be directly applied to dynamic scenes without the risk of Gaussian-object misalignment (i.e., one Gaussian may represent different objects at different time steps). Finally, *Segment then Splat* requires only one pass of reconstruction and does not depend on learning an additional feature field, significantly improving efficiency. In summary, our key contributions include:

- We propose *Segment then Splat*, a novel paradigm that segments Gaussians into object sets before reconstruction. This enables unified *static/dynamic* open-vocabulary segmentation, eliminates auxiliary language fields, and significantly *reduces training complexity*.

- Our framework features a **robust object tracking module** that maintains spatial-temporal consistency of object-specific Gaussians in the scene, ensuring accurate segmentation and motion modeling while preventing misalignment.

- Learning **object-specific Gaussians** from the outset preserves explicit object-Gaussian correspondence, eliminating geometric and semantic ambiguity, yielding superior 3D geometries and multi-level segmentation granularity.

- Extensive experiments demonstrate **state-of-the-art** performance across diverse static and dynamic datasets in 3D open-vocabulary segmentation, object geometry accuracy, and computational efficiency.

## 2 Related Work

### 2.1 3D & 4D Gaussian Splatting

3D Gaussian Splatting (3DGS) [8] is a widely recognized 3D scene representation that introduces anisotropic 3D Gaussians and an efficient differentiable splatting scheme. This enables high-quality explicit scene representation with efficient training and real-time rendering. However, since it was originally designed for static scenes, 3DGS lacks the capability to model dynamic environments.

To address this limitation, Dynamic 3D Gaussians [15] employs a table-based strategy, storing each Gaussian's mean and variance at every timestamp. 4D Gaussian Splatting [16] extends 3DGS into four dimensions, adding a temporal component to facilitate dynamic scene reconstruction. Meanwhile, Deformable 3D Gaussians [17] leverages a multi-layer perceptron (MLP) to learn per-timestamp positions, rotations, and scales for each Gaussian, effectively capturing object motion and deformation over time. 4DGaussians [18] utilizes multi-resolution HexPlanes [19] to decode features for temporal deformation of 3D Gaussians, while STG [20] uses a temporal opacity term and a polynomial function for each Gaussian, yielding a more detailed representation of dynamic scenes.

Despite these advancements, the above methods act solely as scene representations. After reconstruction, they do not support additional interaction or provide information beyond geometry and texture. In contrast, our approach enhances interaction by integrating CLIP [21] embeddings into Gaussian Splatting. This assigns semantic meaning to each Gaussian, enabling open-vocabulary segmentation where users can retrieve, organize, and query objects using natural language prompts.

### 2.2 Language Embedded Scene Representation

Prior to the emergence of radiance field representations, many studies explored the integration of feature embeddings into point cloud representations to enhance scene understanding[22–26]. More recently, a growing body of work has focused on incorporating language features [9, 27, 11, 28, 29] into radiance fields (e.g., NeRF and 3DGS). LERF [9] pioneered this approach by embedding CLIP features in NeRF. Specifically, it extracts pixel-level CLIP embeddings from multi-scale image crops and trains them alongside NeRF to enable open-vocabulary 3D queries. Building on this idea, LEGaussians [11] incorporates uncertainty and semantic feature attributes into each Gaussian, while introducing a quantization strategy to compress high-level language and semantic features. LangSplat [10] employs a scene-wise language autoencoder to learn language features within a scene-specific latent space, and incorporate SAM mask to enable clear object boundaries in rendered feature images. Despite these advancements, all the above methods essentially perform 2D segmentation when conducting open-vocabulary segmentation, as they compare rendered 2D language features with the input text query embedding. OpenGaussian [13] leverages contrastive learning to assign a feature embedding to each Gaussian, then applies $K$-means clustering to group Gaussians into multiple object clusters, thereby realizing 3D segmentation. Similarly, GaussianCut [14] utilizes graphcut [30–32] to segment Gaussians into foreground and background based on user input.

However, all of the above-mentioned methods adhere to a "splat then segment" (i.e., reconstruction then segmentation) pipeline, which inherently results in imprecise object boundaries, since each Gaussian may contain geometry and semantic information from different objects. Furthermore, these methods struggle with dynamic scenes due to Gaussian-object misalignment, preventing their direct application to non-static environments. Dynamic 3D Gaussian Distillation (DGD) [33] distills the feature from LSeg [34] into a feature field to achieve open-vocabulary segmentation. 4D LangSplat [35] further leverages Multimodal Large Language Models (MLLMs) through object-wise video prompting to enhance both time-sensitive and time-agnostic open-vocabulary understanding. Similarly, 4-LEGS [36] distills spatio-temporal language features into 4DGS for event localization from text prompts. Nevertheless, these methods require specific modifications for dynamic scenarios and still suffer from object–Gaussian misalignment.

In contrast, we introduce ***Segment then Splat***, which overturns the long-established "splat then segment" paradigm. Instead, our approach first initializes object-specific Gaussians for each object

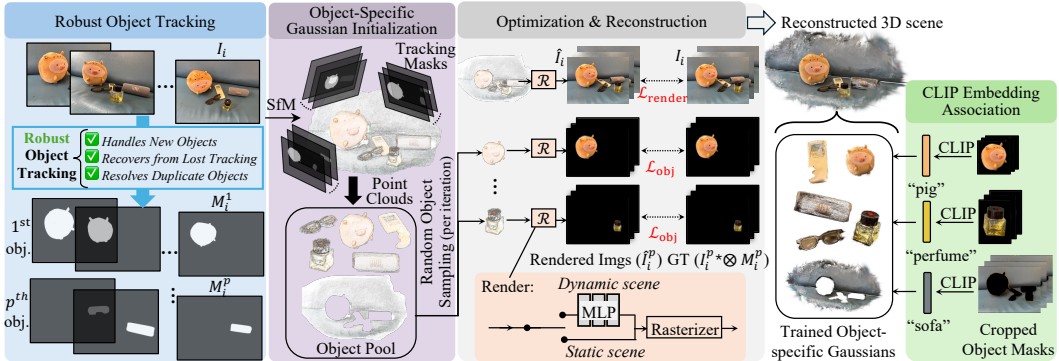

Figure 2: **Demonstration of Segment then Splat pipeline.** We first extract multi-view masks for each object through a robust tracking module, then object IDs are assigned to each initial Gaussian based on these masks, forming distinct object-specific sets. During optimization, object specific loss $\mathcal{L}_{\text{obj}}$ is used to enforce Gaussian-object correspondence and thus resulting in more accurate object geometries. Finally, a CLIP embedding is assigned to each Gaussian group for open-vocabulary queries.

before performing the reconstruction process. By enforcing object-Gaussian correspondence, our method achieves more accurate object geometries and compatibility with dynamic scenes.

## 3 Method

We introduce *Segment then Splat*, a unified approach for 3D open-vocabulary segmentation based on Gaussian Splatting, as illustrated in Fig. 2. The process begins by extracting multi-view masks for each object through a *robust object tracking* module (Sec. 3.2), ensuring reliable detection and mask generation. Next, each Gaussian initialized by COLMAP [37, 38] is assigned an object ID according to these masks, partitioning the entire scene into multiple *object-specific Gaussian sets* (Sec. 3.3). During *optimization & reconstruction* (Sec. 3.4), each Gaussian contributes exclusively to its assigned object, preserving Gaussian-object correspondence and resulting in more accurate object geometries. After reconstruction, *CLIP embeddings are associated* with each group of Gaussians (Sec. 3.5), enabling open-vocabulary queries. Besides, three levels of granularity (large, middle, and small) are introduced to facilitate object retrieval at different scales.

### 3.1 Preliminary: 3D and 4D Gaussian Splatting

**3DGS.** 3D Gaussian Splatting represents a scene using a collection of 3D ellipsoids, each modeled as an anisotropic 3D Gaussian. Each Gaussian is parameterized by a mean $x$, which defines the center of the ellipsoid, a covariance matrix $\Sigma$, which determines its shape, as shown in Eq. (1). The color of the Gaussian is defined using spherical harmonics.

$$G(x) = e^{-\frac{1}{2}(x)^T \Sigma^{-1}(x)}. \tag{1}$$

During rendering, the 3D Gaussians are first transformed into camera coordinates and projected onto the image plane as 2D Gaussians. The final pixel color is then computed through alpha blending, which integrates the weighted Gaussian colors from front to back:

$$\widehat{C} = \sum_{i \in \mathcal{N}} c_i \alpha_i \prod_{j=1}^{i-1} (1 - \alpha_j), \tag{2}$$

where $c_i$ is the color of each Gaussian, and $\alpha_i$ is the alpha value of the $i^{th}$ Gaussian.

**4DGS.** 4D Gaussian Splatting is an extension of 3D Gaussian Splatting, capable of modeling dynamic scenes. Following Deformable 3D Gaussian Splatting [17], we incorporate a deformation field to capture scene dynamics:

$$(\delta \boldsymbol{x}, \delta \boldsymbol{r}, \delta \boldsymbol{s}) = \mathcal{F}_\theta(\gamma(\boldsymbol{x}), \gamma(t)), \tag{3}$$

where $\mathcal{F}_\theta$ represents deformation field, which takes Gaussian mean $\boldsymbol{x}$ and time $t$ as input and outputs the deformation $(\delta \boldsymbol{x}, \delta \boldsymbol{r}, \delta \boldsymbol{s})$ at time $t$. $\gamma(\cdot)$ denotes positional encoding.

## 3.2 Robust Object Tracking

Given a set of input images $\{I_i\}_{i=0}^n$, our goal is to extract multi-view masks for all objects at different granularity levels (i.e., large, middle and small) in the scene. We begin by leveraging Segment Anything (SAM) [39] with grid-based point prompting to obtain initial static object masks of different granularities in the first input frame $I_0$. Subsequently, SAM2 [40] is employed to track objects throughout the sequence based on the extracted mask from $I_0$. However, this process presents several challenges: 1) Some objects may not appear in the first frame, leading to their exclusion from tracking. 2) Due to grid-based prompting, one pixel may be tracked multiple times into different masks, either belonging to a single object or a part of an object. 3) If an object temporarily disappears and reappears later in the scene, tracking may be lost. To overcome these issues and improve the robustness of object tracking, we propose three targeted post-processing strategies that dynamically detect new objects, resolve mask conflicts, and recover from tracking failures. These strategies transform SAM-based segmentation into a flexible, scene-adaptive tracking pipeline, which serves as the critical foundation for subsequent steps in our methodology.

**Detect Any New Objects.** To capture newly appearing objects, we introduce a detection mechanism at fixed intervals of $\Delta t$. Specifically, we compare the segmented region ratio between frames $I_{t+\Delta t}$ and $I_t$. A significant decline in this ratio indicates the potential presence of new objects. At this point, we re-segment the scene and analyze the intersection between the new and previous segmentation results. Objects with minimal overlap with prior masks are identified as new objects and added to the static segmentation results. SAM2 then continues tracking based on this updated segmentation.

**Resolving Multiple Trackings of a Pixel.** To ensure that each pixel is assigned to only one object within a given granularity level, we employ an Intersection over Union (IoU)-based filtering approach. For each pair of masks in $I_i$, if their IoU exceeds a predefined threshold, the smaller object is discarded in favor of the larger one, as it can be segmented separately at a finer granularity level.

**Handling Lost Tracking.** When an object's tracking is lost, it may be incorrectly treated as a new object upon its reappearance in subsequent frames, leading to multiple instances representing the same object. In these scenarios, we resolve this issue using the approach described in Sec. 3.3.

## 3.3 Object-Specific Gaussian Initialization

As our method follows a "segmentation then reconstruction" strategy, we first segment the Gaussians initialized by COLMAP into distinct sets, each representing a different object. Each Gaussian is assigned three object IDs, corresponding to three granularity levels. To determine these IDs, we analyze the visibility of each Gaussian center across all views and identify the corresponding object mask region in which it resides. After object IDs are determined, we handle the lost tracking issue stated in Sec. 3.2. When an object is tracked multiple times as different instances, the corresponding Gaussians should share a similar geometric center and appearance (e.g. color). Therefore, we refine the segmentation by merging Gaussians and its corresponding object mask, if they exhibit a closely aligned "geometric-appearances distance", defined as follows:

$$d(\mathbf{G}_i, \mathbf{G}_j) = \lambda_d |\overline{\mathbf{M}_i} - \overline{\mathbf{M}_j}|_2 + (1 - \lambda_d)|\overline{\mathbf{C}_i} - \overline{\mathbf{C}_j}|_2, \tag{4}$$

where $\mathbf{G}_i$ and $\mathbf{G}_j$ are two sets of Gaussians representing different objects, $\overline{\mathbf{M}_i}$ and $\overline{\mathbf{M}_j}$ denote the mean Gaussian centers, representing object geometric centers, and $\overline{\mathbf{C}_i}$ and $\overline{\mathbf{C}_j}$ are the average colors of the respective Gaussian sets. $\lambda_d$ is the weight to balance geometric distance and appearance distance. Additionally, since COLMAP provides only a sparse reconstruction of the scene, some objects may not be covered and thus lack corresponding Gaussians. We compensate these missing objects by randomly initializing Gaussians. Furthermore, we generate a set of background Gaussians to fill small unsegmented regions.

## 3.4 Optimization & Reconstruction

**Optimization Goal.** After initializing Gaussians as distinct object sets, they are used to reconstruct the scene. During this process, we enforce constraints to ensure that each set of Gaussians contributes only to its corresponding object. Specifically, we introduce an additional object-level loss term,

denoted as $\mathcal{L}_{\text{obj}}$, alongside the standard rendering loss $\mathcal{L}_{\text{render}}$:

$$\mathcal{L}_{\text{render}} = (1 - \lambda_r)\mathcal{L}_1(\hat{I}_i, I_i) + \lambda_r\mathcal{L}_{\text{DSSIM}}(\hat{I}_i, I_i), \tag{5}$$

$$\mathcal{L}_{\text{obj}} = \mathcal{L}_1(M_i^p \otimes I_i, \hat{I}_i^p). \tag{6}$$

where $\hat{I}_i$ is the rendered $i^{\text{th}}$ image, and $\hat{I}_i^p$ and $M_i^p$ represent the rendered $p^{\text{th}}$ object in the $i^{\text{th}}$ image and its corresponding mask, respectively, as described in Sec. 3.2. $\otimes$ denotes element-wise multiplication. The overall loss function is as follows:

$$\mathcal{L} = \mathcal{L}_{\text{render}} + \mathcal{L}_{\text{obj}}. \tag{7}$$

Throughout the entire reconstruction process, all densification and cloning operations are performed strictly within each object-specific Gaussian set, preserving the Gaussian-object correspondence. Besides, a Gaussian persistence mechanism is designed to ensure that each set of Gaussians is not completely pruned, thereby preventing scenarios where an object could lose all its associated Gaussians.

**Optimization Efficiency.** Since the number of objects in a scene can range from a few to over a hundred, computing $\mathcal{L}_{\text{obj}}$ for every object becomes computationally infeasible, as it would require $K$ times more rendering operations per iteration. To address this, we randomly sample $m$ objects per iteration to apply $\mathcal{L}_{\text{obj}}$, balancing efficiency and optimization effectiveness.

**Optimization on Multiple Granularities.** Noticed that we have three level granularities: large, middle and small. To ensure effective optimization, we must account for these varying scales when determining the optimization order. Since smaller objects are always part of larger ones, they should be optimized first. If the order is reversed, i.e., optimizing smaller objects after larger ones, the internal structure of the larger objects may become disorganized again, as shown in Fig. 3. Thus the $\mathcal{L}_{\text{obj}}$ will finally be formulated as:

$$\mathcal{L}_{\text{obj}} = \begin{cases} \mathcal{L}_{\text{obj\_S}}, & \text{stage1} \\ \mathcal{L}_{\text{obj\_S}} + \mathcal{L}_{\text{obj\_M}}, & \text{stage2} \\ \mathcal{L}_{\text{obj\_S}} + \mathcal{L}_{\text{obj\_M}} + \mathcal{L}_{\text{obj\_L}}, & \text{stage3} \end{cases} \tag{8}$$

where $\mathcal{L}_{\text{obj\_S}}$, $\mathcal{L}_{\text{obj\_M}}$, $\mathcal{L}_{\text{obj\_L}}$ represent per-object loss for small, middle, large level respectively.

**Partial Mask Filtering.** In 3D-aware segmentation, objects can be reconstructed even from views where they would typically appear occluded. Consequently, multi-view 2D masks provided as supervision can introduce discrepancies, as they do not account for occluded regions revealed during rendering, as shown in Fig. 5. This discrepancy can lead to incorrect constraints, ultimately distorting the geometric structure of the 3D objects. The original 3DGS framework can partially resolve this issue by leveraging multi-view consistency, but errors from unreliable masks persist. To robustly address this, we propose a partial mask filtering strategy applied at the end of training. Specifically, we render each reconstructed object into 2D images, compute their Intersection-over-Union (IoU) against the provided masks, and discard masks exhibiting low IoU scores. This ensures that only consistent, accurate masks inform the final optimization, significantly enhancing the geometric fidelity of the reconstructed 3D objects.

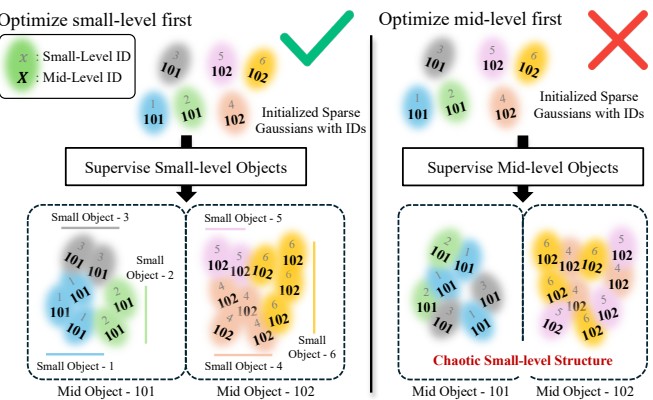

Figure 3: A demonstration of how the optimization order affects reconstruction. Optimizing small-level objects first preserves both small- and middle-level structures, while starting with middle-level ones leads to well-maintained middle-level but chaotic small-level regions due to lack of supervision.

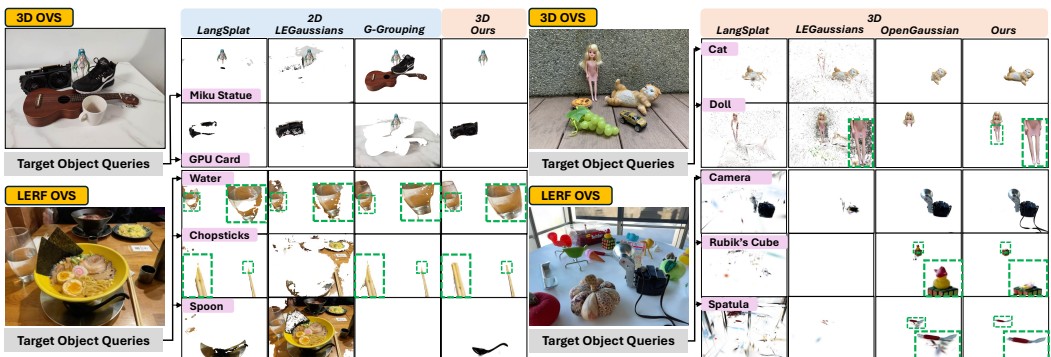

Figure 4: **Qualitative comparison on static scenes.** Compared to baseline methods, our approach accurately retrieves the correct object and produces sharper segmentation boundaries. In contrast, 2D pixel-based methods exhibit ambiguous boundaries, while OpenGaussian either misses parts of the object or incorrectly groups irrelevant objects together.

## 3.5 CLIP Embedding Association

Unlike previous methods [9–12] that supervise Gaussian language embeddings indirectly via 2D feature maps, leading to inconsistent embeddings for Gaussians belonging to the same object, our approach directly assigns a unified language embedding to each object-specific Gaussian set. This ensures consistent semantic embeddings within each object, significantly improving segmentation accuracy and boundary precision. The CLIP embedding of each object is calculated as follows:

$$f_p = \frac{1}{n} \sum_{M_i^p \notin M_{\text{part}}^P} \text{CLIP}_i(\text{crop}(M_i^P \otimes I_i)), \tag{9}$$

where $f_p$ is the language embedding of the $p^{\text{th}}$ object, $M_{\text{part}}^p$ denotes the partial masks that are excluded in the previous section. $\text{CLIP}_i(\cdot)$ represents the CLIP image encoder and $\text{crop}(\cdot)$ denotes the cropping function to extract the mask region.

**Open-vocabulary segmentation.** Given an input text prompt, we perform open vocabulary query following the below strategy:

$$f_q = \text{CLIP}_t(q), \tag{10}$$
$$q_{\text{return}} = \arg\max_p \cos(f_q, f_p), \tag{11}$$

where $f_q$ is the CLIP embedding of the input text prompt $q$, given by CLIP text encoder $\text{CLIP}_t(\cdot)$, and $q_{\text{return}}$ is the object that best matches the query, determined by maximizing the cosine similarity $\cos(\cdot)$ between the query embedding and the object embeddings.

## 4 Experiments

### 4.1 Setups

**Baselines.** We categorize the baselines into two groups based on their querying strategies: *2D pixel-based segmentation* and *3D-based segmentation*. For static scenes, we use LangSplat [10], LEGaussians [11] and Gaussian Grouping [41] as the 2D pixel-based baselines. For the 3D baselines, we choose OpenGaussian[13], and we also adapt LangSplat and LEGaussians for 3D segmentation by selecting Gaussians instead of pixels to evaluate their performance. In dynamic scenes, we adopt a zero-shot image segmentation model CLIP-LSeg [34] as the 2D pixel-based baseline and DGD [33] as the 3D-based approach.

**Datasets & Evaluation Metrics.** To assess the segmentation performance of our proposed method, we conduct experiments on two static scene datasets (i.e., 3DOVS dataset [42] and LERF_OVS dataset [9]) and two dynamic scene datasets (i.e., HyperNeRF dataset [43] and Neu3D dataset [44]). We use mean intersection over union (mIoU) for open-vocabulary segmentation and report optimization time (in minutes) for training efficiency.

**Implementation Details.** The new object detection stride $\Delta t$ in the robust object tracking is set to 10. Following the original 3D Gaussian Splatting, we set $\lambda_r$ in $\mathcal{L}_{\text{render}}$ to 0.2. For geometric-appearances distance, we set $\lambda_d$ to 0.5. In each iteration, we sample 1 object per granularity for 3DOVS to compute $\mathcal{L}_{\text{obj}}$ and 3 objects per granularity for all the remaining datasets. The mIoU threshold for partial mask filtering is set to 30%. We train the smaller-scale 3DOVS data for 20K iterations, and larger-scale datasets (i.e., LERF_OVS, HyperNeRF, and Neu3D) for 40K iterations. All experiments are conducted using a RTX A6000 GPU.

## 4.2 Open-Vocabulary Query

**2D V.S. 3D segmentation.** One thing to be mentioned is that, due to the nature of 3D segmentation and the evaluation used for open-vocabulary segmentation, there is a slight reduction in our mIoU score. This is because 3D segmentation directly retrieves the Gaussians associated with an object, revealing occluded parts that are not visible in some ground truth masks, as shown in Fig. 5.

**Results on 3DOVS dataset.** The quantitative results on the 3DOVS are presented in Tab. 1 (a). Our method outperforms all baseline approaches. The qualitative comparison is shown in Fig. 4. Notably, 2D pixel-based methods tend to produce relatively ambiguous boundaries, whereas our approach, leveraging the ***Segment then Splat*** strategy, achieves significantly clearer object boundaries. For 3D segmentation methods, OpenGaussian requires a predefined number of objects $K$ for clustering. Since 3DOVS contains fewer objects compared to LERF_OVS, we manually reduced its preset $K$. However, even after this adjustment, OpenGaussian still only retrieves parts of certain objects. Although LangSplat and LEGaussians are modified to segment Gaussians instead of pixels, their performance remains suboptimal. This is because each Gaussian's language embedding lacks direct supervision and may encode multiple object semantics, leading to inaccurate segmentation. Additionally, since our method follows a single-pass reconstruction process and the scene scale is relatively small, our training time is significantly shorter compared to the baselines.

Table 1: Quantitative segmentation results for static (a) and dynamic (b) scenes.

**(a) Static scenes**

|  | Method | LERF_OVS mIoU↑ | LERF_OVS Time↓ | 3DOVS mIoU↑ | 3DOVS Time↓ |
|---|---|---|---|---|---|
| 2D | LangSplat [10] | 46.37 | 62.00 | 82.49 | 68.90 |
| | LEGaussians [11] | 18.79 | 72.00 | 52.12 | 55.90 |
| | G-Grouping [41] | 29.59 | 77.00 | 76.24 | 56.10 |
| 3D | LangSplat [10] | 16.76 | 62.00 | 47.31 | 68.90 |
| | LEGaussian [11] | 12.08 | 72.00 | 33.44 | 55.90 |
| | OpenGaussian [13] | 42.43 | 69.75 | 31.00 | 59.40 |
| | **Ours** | **52.10** | **50.75** | **88.53** | **9.40** |

**(b) Dynamic scenes**

|  | Method | HyperNeRF mIoU↑ | HyperNeRF Time↓ | Neu3D mIoU↑ | Neu3D Time↓ |
|---|---|---|---|---|---|
| 2D | LSeg [34] | 15.71 | - | 1.49 | - |
| 3D | DGD [33] | 7.83 | 1564.5 | 1.65 | 1733 |
| | **Ours** | **69.48** | **218** | **44.00** | **161.3** |

**Results on LERF_OVS dataset.** The quantitative results on the LERF_OVS dataset are also presented in Tab. 1 (a) and the qualitative results are shown in Fig. 4. Similar to the 3DOVS dataset, 2D pixel-based methods produce less precise object boundaries, while our method demonstrates significantly improved results. For 3D-based methods, OpenGaussian performs much better on LERF_OVS compared to 3DOVS. However, since OpenGaussian requires a predefined number of clusters, some objects may be incorrectly grouped together (e.g., the rubber duck and the Rubik's cube). Moreover, due to its "splat then segmentation" strategy, its object boundaries remain less accurate than those produced by our method.

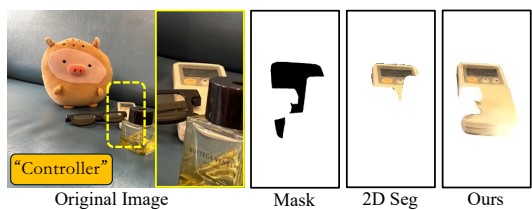

Original Image     Mask     2D Seg     Ours

Figure 5: Comparison between 2D pixel-based segmentation and our 3D segmentation. Unlike 2D pixel-based methods, which are limited by occlusions, our approach can retrieve the complete object even from an occluded view.

**Results on HyperNeRF dataset.** The quantitative results and qualitative results on HyperNeRF dataset are presented in Tab. 1 (b) and Fig. 6 respectively. Since our method explicitly enforces Gaussian-object correspondence, it can be directly applied to dynamic scenes, achieving good segmentation performance without the Gaussian-object misalignment issue encountered by previous approaches. In contrast, methods such as DGD, which relies on learning a language field supervised

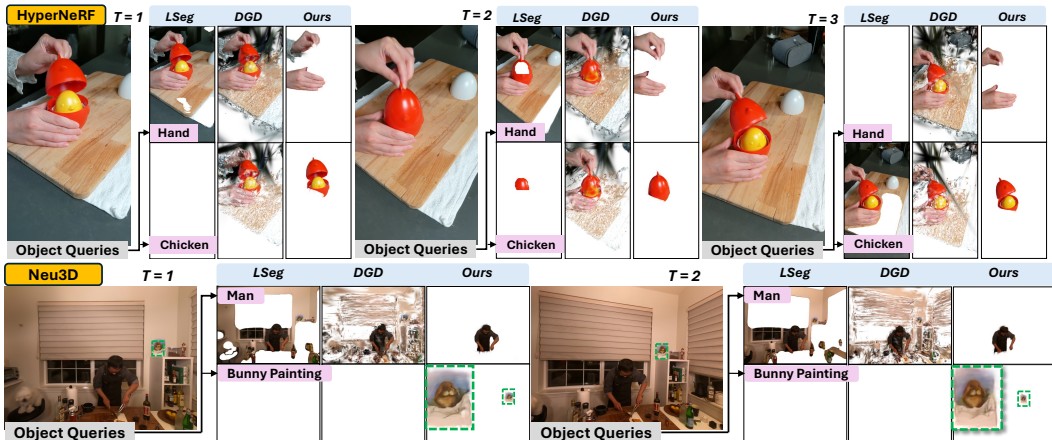

Figure 6: **Qualitative comparison on dynamic scenes.** As our method enforce object-Gaussian correspondence, it applies directly to dynamic scenes and performs well, whereas DGD and LSeg tend to include irrelevant content and are not able to retrieve small objects (e.g., bunny painting).

via 2D feature maps, suffer from misalignment, as a single Gaussian might represent multiple distinct objects across different time steps. As illustrated in Fig. 6, this misalignment results in retrieving irrelevant Gaussians. Moreover, because DGD does not directly supervise the language embeddings of each Gaussian, Gaussians located far apart may share similar embeddings, further deteriorating segmentation quality. In addition, our method achieves nearly a ten-fold improvement in optimization speed compared to DGD, as learning a dynamic language field is computationally intensive. We omit training time results for LSeg, as it is a zero-shot method requiring no additional optimization.

**Results on Neu3D dataset.**
Quantitative results and qualitative results are shown in Tab. 1 and Fig. 6. Similar to the observations on the HyperNeRF dataset, many irrelevant Gaussians are retrieved due to object-Gaussian misalignment issue and the "splat then segment" strategy. Besides, both LSeg and DGD fail when retrieving relatively small objects (e.g., bunny painting).

Table 2: Ablation studies: (Top) Number of supervised objects per iteration. (Middle) Partial mask filtering. (Bottom) Tracking module ablation.

**(a) Number of supervised objects per iteration**

| | Static Scene | | | | Dynamic Scene | | | |
| | ramen | | waldo_kitchen | | chickchicken | | split-cookie | |
| # obj | Time↓ | mIoU↑ | Time↓ | mIoU↑ | Time↓ | mIoU↑ | Time↓ | mIoU↑ |
|---|---|---|---|---|---|---|---|---|
| 1 | 26 | 51.09 | 44 | 33.97 | 146 | 73.11 | 180 | 77.76 |
| 3 | 34 | 54.38 | 48 | 40.71 | 157 | 74.89 | 202 | 80.30 |
| 5 | 42 | 55.61 | 54 | 40.83 | 172 | 75.29 | 253 | 80.47 |
| 7 | 53 | 56.03 | 63 | 40.77 | 185 | 75.21 | 258 | 81.52 |
| 9 | 62 | 56.48 | 68 | 41.59 | 201 | 75.23 | 309 | 81.74 |

**(b) Partial mask filtering strategy**

| | Static Scene | | | | Dynamic Scene | | | |
| | ramen | | waldo_kitchen | | chickchicken | | split-cookie | |
| Method | mIoU↑ | PSNR↑ | mIoU↑ | PSNR↑ | mIoU↑ | PSNR↑ | mIoU↑ | PSNR↑ |
|---|---|---|---|---|---|---|---|---|
| w/o MF | 42.19 | 23.11 | 31.94 | 30.43 | 72.65 | 29.32 | 80.23 | 32.76 |
| **Ours** | **54.38** | **24.54** | **40.71** | **31.33** | **74.89** | **29.62** | **80.30** | **33.04** |

**(c) Tracking module ablation**

| Method | ramen | | teatime | | figurines | | split-cookie | |
| | ORR↑ | Dup↓ | ORR↑ | Dup↓ | ORR↑ | Dup↓ | ORR↑ | Dup↓ |
|---|---|---|---|---|---|---|---|---|
| SAM2 | 0.889 | 1 | 0.858 | 1 | 0.683 | 4 | 0.970 | 0 |
| +new_obj_detect | 0.934 | 2 | 0.961 | 3 | **0.963** | 18 | 1.000 | 2 |
| +multi-track | 0.934 | 1 | 0.961 | 3 | 0.948 | 5 | 1.000 | 0 |
| +lost-track | **0.934** | **0** | **0.961** | **0** | 0.948 | **0** | **1.000** | **0** |

## 4.3 Ablation Study

**Number of Supervised Objects.** In this ablation study, we examine the relationship between the number of supervised objects in each granularity per iteration and the segmentation performance. We pick two static scenes from LERF_OVS and two dynamic scenes from HyperNeRF dataset. The results are presented in Tab. 2 (a). As expected, increasing the number of supervised objects per iteration generally enhances segmentation performance. However, beyond a certain threshold, the performance gain becomes marginal. For scenes with fewer objects (e.g., chickchicken and split-cookie), performance quickly converges after a certain number of supervised objects. In contrast, more complex scenes containing many objects (e.g., ramen and waldo_kitchen) continue to show performance improvements, though at a diminished rate. Additionally, training time increases

proportionally with the number of supervised objects. To balance training time and performance, we choose three objects for LERF_OVS, Neu3D and HyperNeRF in our experiments.

**Partial Mask Filtering.** In this ablation study, we investigate the impact of the partial mask filtering strategy on segmentation performance as well as reconstruction quality. As shown in Tab. 2 (b), scenes with a larger number of objects (e.g., ramen, waldo_kitchen) tend to have more occluded views, leading to more performance gain when applying partial mask filtering. In contrast, scenes with fewer objects (e.g., chickchicken, split-cookie) exhibit a smaller performance gain. Overall, the results validate the effectiveness of the partial mask filtering strategy, particularly in complex scenes with high object density.

**Robust Object Tracking.** We conduct an ablation study on each component of our robust object tracking module, as shown in Tab. 2 (c). The evaluation is performed on three static scenes from LERF_OVS and one dynamic scene from HyperNeRF. Leveraging ground-truth labels, we adopt two metrics: Object Recall Rate (ORR), defined as

$$\text{ORR} = \frac{1}{k} \sum_{i=1}^{k} \frac{\text{number of tracked objects}}{\text{number of GT objects}}, \tag{12}$$

where k is the number of ground-truth frames, measures the percentage of objects successfully tracked throughout the sequence; and the number of duplicate trackings (Dup), which quantifies how often the same object is redundantly tracked as multiple instances. Results show that the new object detection module increases ORR by capturing newly appeared objects, albeit at the cost of more duplicate instances. In contrast, the multi-track and lost-track handling modules effectively reduce Dup, as described in Sec. 3.2. The slight drop in ORR observed in the Figurines scene when adding the multi-track module is caused by an isolated tracking failure at a fine-grained level in a single frame. However, this minor failure does not affect the final reconstruction, as sufficient information is retained from other views.

## 5 Conclusion

We propose *Segment then Splat*, a unified framework for 3D open-vocabulary segmentation based on Gaussian Splatting. We reverse the long-established "segment after reconstruct" approach to form a "segment then reconstruct" pipeline. By maintaining consistent object–Gaussian correspondence throughout the process, *Segment then Splat* effectively eliminates both geometric and semantic ambiguities, resulting in more precise object boundaries and improved segmentation performance. Furthermore, because this correspondence is explicitly established, our method naturally extends to dynamic scenes without concerns of misalignment between objects and Gaussians during dynamic modeling. Extensive experiments across diverse static and dynamic datasets demonstrate the superior performance and robustness of our framework.

## 6 Limitation and Future Direction

Our method relies on SAM2 for initializing object tracking. In extremely complex scenarios with high-density and visually similar objects, tracking may fail, resulting in suboptimal open-vocabulary segmentation, which is an issue commonly shared by tracking-based approaches (e.g., Gaussian Grouping). This limitation could potentially be addressed by incorporating techniques such as Kalman filtering [45] to improve the temporal stability of SAM2. Another limitation is that our method cannot effectively handle text queries involving relational descriptions across multiple objects, such as "a sheep sitting on the chair in front of the table." Similar to existing baselines, our method encodes textual embeddings from masked regions or local patches corresponding to individual objects, and therefore lacks explicit modeling of inter-object relationships or contextual understanding. Developing this capability is crucial for advancing open-vocabulary understanding, and we plan to explore it in future work.

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
