# OpenReview forum: "Segment then Splat: Unified 3D Open-Vocabulary Segmentation via Gaussian Splatting"
_NeurIPS.cc/2025/Conference — NeurIPS 2025 poster_

### Official Review · Reviewer_cG1h · 2025-06-24

**Clarity:** 3
**Significance:** 2
**Originality:** 2
**Rating:** 3
**Confidence:** 4

**Summary:**

This paper proposes ‘segment then splat’, a 3D open-vocabulary scene understanding method under the framework of 3D-GS. Compared with previous semantics reconstruction then understanding methods like LangSplat and LEGaussians, the proposed method first segments the whole scene into multiple objects and attach semantics information to each object. Experiments results demonstrate the effectiveness of the proposed method.

**Questions:**

1. I think it is better to focus on the segmentation task like GaussianGrouping. Though the paper claims it focuses on open-vocabulary segmentation, the main method focuses on improving the class-agnostic segmentation.
2. My main concern is about the novelty, which I think can not be solved by minimal modification to the current paper. I encourage the author to totally revise the paper and method and resubmit.

**Ethical Concerns:**

["NO or VERY MINOR ethics concerns only"]

**Final Justification:**

I thank the authors for their response. After reading the response I still believe the method focuses on class-agnostic 3D segmentation. I think it requires a more holistic way to integrate the semantic understanding part with the segmentation part. Considering the technical contribution of the paper, I raise my score from 2 to 3.

**Limitations:**

Yes.

**Quality:**

2

**Strengths And Weaknesses:**

Strengths:

1. The motivation and the proposed method are easy to understand.
2. Writing is clear.
3. Experimental results are good, showing the effectiveness of the proposed method.

Weaknesses:
1. There is a misalignment between the theme of the paper and the proposed method. Though the paper claims the proposed method focuses on open-vocabulary 3D segmentation, the main part of the proposed method focuses on segment the 3D scene in a class-agnostic paradigm (similar to Gaussian Grouping). The semantic understanding is achieved by simple multi-view semantics feature aggregation.
2. The whole pipeline is not new. The paradigm of Segment then Splat’ has been proposed by OpenGaussian. Discussion is insufficient. The only new thing is the support for dynamic scene, which is not a core challenge of open-vocabulary 3D understanding.
3. How to obtain the 42.43 mIoU of OpenGaussian in Table 1? The mIoU reported in their paper is 38.36.
4. Many important citations are missing, including but not limited to N2F2 [1], SAGA [2], GARField [3], Feature3DGS [4], GOI [5], and others. I strongly recommend that the authors conduct a more thorough review of related work in this field.
5. The main contribution of the proposed method is unclear. The 3D segmentation method looks like a collection of tricks (Line 148-160) added on GaussianGrouping. However, the reliability of such hand-crafted calibration is in doubt. As shown in Table 2, these tricks do not provide large performance gain.

[1] Bhalgat, Y., Laina, I., Henriques, J. F., Zisserman, A., & Vedaldi, A. (2024, September). N2f2: Hierarchical scene understanding with nested neural feature fields. In European Conference on Computer Vision (pp. 197-214). Cham: Springer Nature Switzerland.
[2] Cen, J., Fang, J., Yang, C., Xie, L., Zhang, X., Shen, W., & Tian, Q. (2025, April). Segment any 3d gaussians. In Proceedings of the AAAI Conference on Artificial Intelligence (Vol. 39, No. 2, pp. 1971-1979).
[3] Kim, C. M., Wu, M., Kerr, J., Goldberg, K., Tancik, M., & Kanazawa, A. (2024). Garfield: Group anything with radiance fields. In Proceedings of the IEEE/CVF Conference on Computer Vision and Pattern Recognition (pp. 21530-21539).
[4] Zhou, S., Chang, H., Jiang, S., Fan, Z., Zhu, Z., Xu, D., ... & Kadambi, A. (2024). Feature 3dgs: Supercharging 3d gaussian splatting to enable distilled feature fields. In Proceedings of the IEEE/CVF Conference on Computer Vision and Pattern Recognition (pp. 21676-21685).
[5] Qu, Y., Dai, S., Li, X., Lin, J., Cao, L., Zhang, S., & Ji, R. (2024, October). Goi: Find 3d gaussians of interest with an optimizable open-vocabulary semantic-space hyperplane. In Proceedings of the 32nd ACM International Conference on Multimedia (pp. 5328-5337).

---

> ### Author Rebuttal · Authors · 2025-07-31
>
> We thank the reviewer for recognizing the effectiveness of our method and providing feedbacks. We would like to take this opportunity to clarify the following points.
>
> # \[W1, Q1\] Clarification of Motivation and Contribution
>
> Thank you for your feedback. While we appreciate the reviewer’s perspective, we would like to respectfully clarify our motivation and contributions. Our primary focus is on open-vocabulary segmentation, and our core motivation is to assign each Gaussian an accurate and discriminative language embedding to enable precise text-driven segmentation. We achieve this by eliminating semantic ambiguities and establishing correspondence between Gaussian groups and objects (e.g., Gaussians belonging to the same object have identical embeddings and each Gaussian only represents one object).
>
> Specifically, we design a robust tracking module and develop Gaussian-specific optimization strategy, all these are served for establishing object-Gaussian correspondence and eliminating ambiguity. These components are central to our method and serve the overall goal of improving open-vocabulary segmentation performance.
>
> Therefore, we respectfully disagree with the assessment that our method primarily focuses on class-agnostic segmentation. While class-agnostic grouping is used as an intermediate step, our full pipeline is thematically and technically aligned with the goal of open-vocabulary 3D segmentation.
>
> # \[W2, W5, Q2\] Novelty
>
> We would like to emphasize the novelty and contributions of our proposed pipeline, and clarify how it fundamentally differs from previous works.
>
> All prior methods (e.g., OpenGaussian, LangSplat, Gaussian Grouping, etc.) follow a “Splat-then-Segment” paradigm. Taking OpenGaussian (mentioned by the reviewer) as an example, it first **reconstructs** the scene using 3D Gaussian Splatting while simultaneously learning a feature field via contrastive loss. It then clusters the Gaussians based on the learned features, and finally assigns CLIP embeddings to each cluster to enable open-vocabulary **segmentation**.
>
> “Splat-then-Segment” will cause several issues.
>
> 1. **Geometric and Semantic Ambiguities:** Without object-level constraints during reconstruction, Gaussians near object boundaries may represent multiple objects and carry mixed semantic information, leading to ambiguity.
> 2. **Object-Gaussian Misalignment in Dynamic Scenes**: In dynamic scenes, a single Gaussian may represent different objects across time, requiring impractical time-varying embeddings for open-vocabulary segmentation. Thus, direct adaptation of “Splat-then-Segment” methods to dynamic scenes is fundamentally limited.
>
> We propose Segment-then-Splat, we first “segment” Gaussians into object-specific groups then during reconstruction, each group of Gaussians only contributes to its own object, thus eliminating the ambiguity and misalignment issue inherent in “Splat-then-Segment” approaches.
>  Our method achieves superior segmentation performance compared to baselines, particularly around object boundaries (see Fig. 4 in the main paper), and can be directly applied to dynamic scenes without additional modification.
>
> # \[W3\] Higher mIoU of OpenGaussian compared to the original paper
>
> Thank you for your observation. The original OpenGaussian implementation holds out every eighth image (i.e., llffhold=8) during training to follow the standard evaluation protocol for reconstruction. However, this results in some segmentation-labeled images being inadvertently included in the training set, which is inconsistent with the evaluation protocols followed by other baseline methods.
> To ensure a fairer evaluation and to align with the settings used by other baseline methods, in segmentation experiments, we exclude all labeled images from the training set and use only the unlabeled images for training. This adjustment allows more images to be used for reconstruction compared to the llffhold=8 setting,  thus leading to higher reconstruction quality and, consequently, improved segmentation performance.
>
> # \[W4\] Missing Citations
>
> Thank you for pointing this out. As some of the referenced works are not directly focused on open-vocabulary segmentation, we did not include them primarily in the current version. However, we acknowledge their relevance and will carefully review these works and consider including appropriate citations and discussions in the final version of the paper.
>
> # \[W5\] Reliability of the Robust Tracking Module
>
> We have conducted comprehensive experiments on 4 datasets including LERF-OVS, 3DOVS, Neu3D, and HyperNeRF ensuring its robustness. Additionally, we conduct new experiment on the more challenging MipNeRF360 dataset, including scene with hundreds of cluttered objects (e.g., *Counter* and *Bonsai*) to showcase the reliability and effectiveness of our method, the results are provided below:
>
> | Method        | Counter mIoU | Time | Bonsai mIoU | Time | Kitchen mIoU | Time | Garden mIoU | Time | Room mIoU | Time | Avg mIoU | Avg Time |
> |---------------|--------------|------|-------------|------|---------------|------|--------------|------|------------|------|----------|-----------|
> | Ours          | **55.32**    |**36.0**|**63.80**    |**45.0**|**68.59**      |**48.0**|**44.09**      |**86.0**| 42.90      | 44.0 |**54.94** |**51.80**   |
> | OpenGaussian  | 49.57        | 48.0 | 39.08       | 58.0 | 31.22         | 60.0 | 15.10        | 130.0| 41.88      |**42.0**| 35.37    | 67.60     |
> | G-Grouping    | 41.76        | 50.0 | 56.36       | 62.0 | 58.15         | 71.0 | 36.25        | 128.0| 30.84      | 63.0 | 44.67    | 74.80     |
> | LEGaussian    | 54.98        | 50.0 | 62.46       | 52.0 | 66.77         | 60.0 | 40.12        | 103.0|**45.58**    | 61.0 | 53.98    | 65.20     |

---

> > ### Comment · Reviewer_cG1h · 2025-08-05
> >
> > Thank the authors for their response. After reading the response I still believe the method focuses on class-agnostic 3D segmentation. I think it requires a more holistic way to integrate the semantic understanding part with the segmentation part.

---

> > > ### Comment · Area_Chair_74ht · 2025-08-06
> > > **please provide a more detailed answer**
> > >
> > > Dear cG1h (authors are included on this one),
> > >
> > > Given that three of the reviewers lean positive, and acknowledge that the method is open-vocabulary segmentation, we would need a more detailed explanation of your assessment. Do the rest of the authors' clarifications & experiments satisfy your concerns?
> > >
> > > Thank you
> > > AC

---

> > > > ### Comment · Reviewer_cG1h · 2025-08-06
> > > >
> > > > Dear AC and authors,
> > > >
> > > > I'm willing to further clarify my judgement. I have no other concerns about the paper except the novelty and technical contribution. The method and experiment results are general okay.
> > > >
> > > > However, the whole pipeline 'Segment-then-splat' has been proposed by an existing method (OpenGaussian). Though in the author response, they try to clarify the contribution and novelty of the proposed method. I think the response is not convincing enough.
> > > >
> > > > Especially, the response to [W2, W5, Q2] Novelty, which says OpenGaussian follows a Splat then Segment pipeline, is not accurate. OpenGaussian first learns contrastive features to decompose the whole 3D scene into multiple objects and then assign semantic features to each cluster. I think this is a typical paradigm of 'Segment then splat'.
> > > >
> > > > It is worth note that the main illustration of this paper (Figure 1) is also very similar to OpenGaussian (also Figure 1), both introducing the difference between previous 'splat-then-segment' paradigm and the newly introduced 'segment-then-splat' paradigm. Thus I respect but cannot agree with the author's response.
> > > >
> > > > Moreover, using object tracking module for 3D-GS decomposition is also not a new thing, which is first introduced in GaussianGrouping and followed by many other methods.
> > > >
> > > > The paper does introduce new techniques for improvement (all focusing on the segment phase), but its claimed core contribution is untenable. As evidenced by Figure 2 and 3, all illustrations are about class-agnostic segmentation. The semantic assignment process is only briefly described (Line 224-237) and uses a very traditional approach. Thus, I hope the authors can totally revise the paper and mainly focus on the class-agnostic segmentation task.

---

> ### Author Response · Authors · 2025-08-06
>
> Dear AC and Reviewers,
>
> We thank AC for initiating this discussion, and also appreciate reviewer cG1h for his/her valuable feedback.
> 1. First we want to justify why OpenGaussian is not “Segment-then-Splat”.
> a. The following are the training steps stated in the official github repository of OpenGaussian:
>
> ```
> [Stage 0] Start 3dgs pre-train ... (step 0-30k)
> [Stage 1] Start continuous instance feature learning ... (step 30-50k)
> [Stage 2.1] Start coarse-level codebook discretization ... (step 50-70k)
> [Stage 2.2] Start fine-level codebook discretization ... (step 70-90k)
> [Stage 3] Start 2D language feature - 3D cluster association ... (1 min)
> ```
>
>
> OpenGaussian first trains 3DGS for 30k steps in stage 0, which is “Splat”, then in the stage1 it utilizes contrastive learning to learn a feature field. After that in stage 2.1 and 2.2 it conducts codebook discretization, which clusters Gaussians into different groups, this is “Segment”. Thus, OpenGaussian is “Splat-then-Segment”, not “Segment-then-Splat”.
>
> b. Similar evidence can also be found in the appendix A.1(1) section of the OpenGaussian paper. They claim their training strategy is consistent with LangSplat (also Splat-then-Segment), we copy and paste the paragraph here:
>
> *Training Strategy. Consistent with LangSplat, we first pre-train the standard 3DGS for 30,000*
> *steps. Subsequently, we freeze the Gaussian coordinates, scale, and opacity parameters, and train the*
> *instance features for 10,000 steps (ScanNet is 20,000 steps) and the two-layer codebook for 30,000*
> *steps (ScanNet is 40,000 steps). The 2D-3D feature association step is training-free. The extraction*
> *methods for SAM masks and CLIP features also align with LangSplat. While LangSplat extracts*
> *three layers of SAM masks (small, middle, and large), our implementation uses only one layer (large).*
>
> c. This distinction also explains why OpenGaussian still suffers from geometric and semantic ambiguities. Please refer to the supplementary video material for a direct segmentation comparison between our method and OpenGaussian. As demonstrated, our method produces notably more accurate and precise object boundaries than OpenGaussian.
>
> 2\. Regarding the perceived similarity between our Fig. 1 and OpenGaussian's Fig. 1\. The differences are as follows:
>
> - We are comparing the difference between Splat-then-Segment  and Segment-then-Splat. While OpenGaussian is comparing 2D segmentation and 3D segmentation, which is totally different.
> - We include both training and inference stage in the figure, while they only include the inference stage.
>
> The reviewer thinks they are similar, maybe because in Fig.1(a) we are demonstrating the splat-then-segment pipeline, which OpenGaussian follows.
>
> 3.Furthermore, we would like to clarify the novelty of our proposed tracking module, which fundamentally differs from that employed by GaussianGrouping. GaussianGrouping independently performs per-frame segmentation first and then employs a temporal propagation model to associate masks across frames. This approach inherently risks breaking multiview consistency, as it assumes that segmentation of the same object remains identical across frames. For instance, an object segmented as a single instance in one frame (e.g., a man wearing a hat) might be segmented as separate instances (e.g., a man and a hat) in a subsequent frame. Indeed, if one examines the project page or GitHub repository of GaussianGrouping, the provided GIFs or videos clearly illustrate frequent abrupt mask changes, directly evidencing compromised multiview consistency.
>
> In contrast, our tracking module leverages the mask from the first frame as an initial prompt and employs SAM2 to propagate this mask consistently across subsequent frames, thus explicitly ensuring temporal consistency. Simultaneously, our robust tracking strategy effectively detects newly appearing objects and manages issues such as tracking loss. This methodology fundamentally differs from the tracking approach employed by GaussianGrouping.
>
> Moreover, GaussianGrouping does not explicitly utilize the tracked masks to group Gaussians. Instead, it learns an implicit identity encoding through differentiable rendering, which further distinguishes their method from ours.
>
> 4\. We thank the reviewer for recognizing the new techniques we introduced, however, as we stated in rebuttal \[W1, Q1\], all these designs are primarily focusing on how to eliminate the geometry and semantic ambiguity of the Gaussian, thus enabling a better performance in open-vocabulary segmentation. We respect the perspective of the reviewer, but don’t agree that our work is focusing on class-agnostic segmentation.

---

> > ### Comment · Reviewer_cG1h · 2025-08-06
> >
> > Dear authors,
> >
> > Thanks for the detailed explanation, which further explains the difference between Segment-then-splat and OpenGaussian. Now I can understand the disagreement between the authors and me.
> >
> > The proposed method integrates the segmentation phase into the reconstruction of the 3D-GS scene, while OpenGaussian first reconstructs the whole scene and then conducts class-agnostic segmentation. After the segmentation (i.e., segmentation with reconstruction in Segment-then-splat) phase, both methods assign semantic information to the class-agnostic objects. The authors believe that this design makes Segment-then-splat pipeline different from OpenGaussian.
> >
> > However, from the perspective of Open-vocabulary segmentation, both methods follow a same paradigm that **first segmenting the whole 3D scene and then adopting CLIP to assign semantics to each object in it**. Thus, this makes me believe Segment-then-splat is essentially same with OpenGaussian (the only difference and improvement is **the way to segment the whole scene in a class-agnostic way**).
> >
> > Regarding Figure 1, the authors claim that 'We are comparing the difference between Splat-then-Segment and Segment-then-Splat'. However, in Figure 1, an important characteristic of Splat-then-segment is they conduct 'Pixel-level Segment'. However, **OpenGaussian conducts 3D Gaussians level segmentation**, which is the key contribution of it.
> >
> > I agree that this paper does have improvement compared with previous methods, and I understand that the motivation behind such modification may root in open-vocabulary segmentation. But the main contribution of this paper lies in finer-grained class-agnostic segmentation (object-aware 3D scene reconstruction).

---

> > > ### Author Response · Authors · 2025-08-06
> > >
> > > We thank the reviewer for recognizing that OpenGaussian is not “Segment-then-Splat”, but “Splat-then-Segment”. This demonstrates that our method and OpenGaussian are **logically different**.
> > >
> > > Our method follows: **pre-segment 3D Gaussians** \-\> **instance-level reconstruction** \-\> **assign CLIP embedding to each instance.**
> > > OpenGaussian follows: **scene-level reconstruction** \-\> **Gaussian clustering** \-\> **assign CLIP embedding to each cluster.**
> > >
> > > However, we disagree with the reviewer’s assessment that the two methods are essentially the same in the perspective of open-vocabulary segmentation. The reviewer intended to merge the first two steps to **segment the whole 3D scene** to state that the two methods are the same. ​​We respectfully believe that the reviewer’s abstraction, while reasonable at a very high level, risks oversimplifying and flattening substantial technical differences across methods.
> > >
> > > Here, we list the key differences between our proposed “Segment-then-Splat” and OpenGaussian below:
> > >
> > > 1. In OpenGaussian, segmentation is applied post hoc on a pre-trained, 3D Gaussian scene, where the Gaussians are not object-aware during reconstruction. This delayed segmentation will lead to **geometric and semantic ambiguities**. In contrast, our method segments Gaussians into groups ahead of reconstruction, to ensure an object-level reconstruction that improves consistency between geometry and semantics.
> > > 2. OpenGaussian requires jointly training a dense feature field alongside the 3D Gaussian Splatting reconstruction, which requires additional computational cost. In contrast, our method eliminates the need for feature field training, resulting in a more efficient pipeline. On the LERF dataset, our method **reduces training time by approximately 30%** compared to OpenGaussian, while **maintaining superior performance**.
> > > 3. OpenGaussian **cannot be directly applied to dynamic scenes**, as its clustering strategy, the codebook discretization, relies on a static 3D scene representation. In dynamic settings, the relationship between Gaussians and objects evolves over time, making it challenging to maintain consistent and meaningful cluster assignments. In contrast, our method can be directly applied to dynamic scenes without any modification, as it establishes object-Gaussian correspondence in the first stage.
> > >
> > > Given the listed difference above, we believe that our proposed “Segment-then-Splat” is **fundamentally different** from OpenGaussian in terms of pipeline design, compatibility with dynamic scenes, and training efficiency.
> > >
> > > ## Figure 1
> > >
> > > We apologize for the confusion introduced by Fig.1. In Figure 1(a), we demonstrate the most common “Splat-then-Segment” pipeline, as most of the existing open-vocabulary segmentation methods are “pixel-based”. This does not mean “pixel-based” segmentation is an important characteristic of “Splat-then-Segment”. Instead, as their name indicates, the key difference between “Segment-then-Splat” and “Splat-then-Segment” is the order of reconstruction (blue bounding box in Fig.1) and segmentation (orange bounding box in Fig.1). And OpenGaussian follows the operation flow demonstrated in Fig. 1(a), which indicates it is “Splat-then-Segment”.
> > > We thank the reviewer for pointing this out, we will refine the Fig.1 in the revised version to eliminate confusion.
> > >
> > > ## Open-vocabulary Segmentation or Class-agnostic segmentation
> > >
> > > We respect the reviewer’s perspective but respectfully disagree with the characterization that our main contribution lies in finer-grained class-agnostic segmentation. To illustrate this point with an analogy: if one designs a new backbone architecture for an object detection system, while keeping the detection head unchanged. The contribution is still to object detection, not merely to backbone design.
> > >
> > > In our case, our “Segment-then-Splat” method serves as a new *backbone* specifically designed to overcome the challenges inherent in open-vocabulary segmentation, as stated in rebuttal \[W1,Q1\]. Its effectiveness is therefore defined and measured by its performance on the end-to-end task of open-vocabulary segmentation, which is the central focus of our experiments and analysis.

---

> > > > ### Comment · Reviewer_cG1h · 2025-08-08
> > > >
> > > > Thanks for the detailed explanation. Considering the technical contribution of this paper, I would like to raise my score from 2 to 3. That said, my recommendation remains Weak Reject for the following reasons.
> > > >
> > > > First, geometric and semantic ambiguities are not specific to open-vocabulary segmentation; they also arise in class-agnostic segmentation, as noted in prior work [1]. This weakens the paper’s claim that its core motivation uniquely stems from the open-vocabulary setting.
> > > >
> > > > Second, if the goal is truly open-vocabulary segmentation, the integration between segmentation/reconstruction and semantic classification needs to be tighter. In the current pipeline, **the segmentation/reconstruction stage does not incorporate any CLIP semantics**, which raises two questions: (i) how is the granularity of CLIP semantics aligned with the class-agnostic segmentation output? (over- and under-segmentation cause semantic ambiguity) and (ii) if 3D scenes are pre-segmented and semantics are assigned post hoc via CLIP on 2D masked images, how are inter-object relationships—an essential part of scene semantics—captured?
> > > >
> > > > In conclusion, while the paper positions the method as tailored to open-vocabulary segmentation, the design appears largely class-agnostic. Removing the CLIP-based head essentially yields a class-agnostic segmentation model. A more accurate—and, in my view, stronger—framing would be: the paper proposes a better class-agnostic segmentation approach whose object-aware design also benefits open-vocabulary segmentation.
> > > >
> > > > By the way, the authors’ analogy to “a new backbone for detection” is not apt: in detection, the backbone is trained end-to-end under semantic supervision from the head, whereas here the segmentation/reconstruction stage is class-agnostic and receives no semantic signal; functionally, it acts as a decoupled pre-processing step rather than a semantics-aware backbone.
> > > >
> > > > [1] Hu, X., Wang, Y., Fan, L., Fan, J., Peng, J., Lei, Z., … Zhang, Z. (2024). SAGD: Boundary-enhanced segment anything in 3D Gaussian via Gaussian decomposition. arXiv:2401.17857.

---

> > > > > ### Author Response · Authors · 2025-08-09
> > > > >
> > > > > We thank the reviewer for recognizing the technical contributions of our paper and sincerely appreciate the raised score. While we respectfully disagree with the reviewer’s opinion that this work primarily addresses class-agnostic segmentation, we understand and acknowledge that this may stem from differing perspectives. We truly value the thoughtful exchange during the discussion period and are grateful for the reviewer’s time, patience, and constructive feedback. Below, we address the reviewer’s questions and comments.
> > > > >
> > > > > ## Geometric and semantic ambiguities
> > > > >
> > > > > We agree with the reviewer’s statement that geometric and semantic ambiguities are not specific to open-vocabulary, but it is still one of the crucial problems in open-vocabulary segmentation. With the existence of these ambiguities, the language embedding can not be accurately assigned to each Gaussian, leading to incorrect Gaussian retrieval and, consequently, degraded open-vocabulary segmentation performance.
> > > > >
> > > > > ## Alignment of CLIP embedding and segmentation output
> > > > >
> > > > > In our proposed method, each Gaussian is associated with object IDs at three different granularity levels. Consequently, we obtain three levels of Gaussian segmentation results, along with their corresponding multiview masks, which are generated by our robust tracking module. For each instance at each granularity level, we compute a multiview-consistent CLIP embedding using its associated multiview masks. This embedding is then assigned to all Gaussians that comprise the object. As a result, each Gaussian carries three embeddings, capturing its semantic meaning at the whole-object, part, and sub-part levels.
> > > > >
> > > > > ## Inter-object relationship
> > > > >
> > > > > We thank the reviewer for highlighting this point. Indeed, one limitation of our approach is that the language embeddings are derived solely from masked object images, meaning they capture only the semantics of individual objects and do not account for contextual relationships between different objects. However, we would like to emphasize that **this limitation is shared by most existing open-vocabulary segmentation methods** (e.g. LERF, LEGaussian, LangSplat, etc.). We agree that incorporating inter-object contextual understanding will be a promising direction for future research.

---

### Official Review · Reviewer_NzNg · 2025-06-28

**Clarity:** 2
**Significance:** 2
**Originality:** 2
**Rating:** 4
**Confidence:** 4

**Summary:**

This submission proposes a heuristic approach to segment (potentially dynamic) scenes observed in multiple images and assign CLIP descriptors to them.

**Questions:**

How are the 3 scales determined?  Does the approach assume a know metric scale?  How does the approach work for a camera moving through a scene which can observe objects both from close-by and far away?

**Ethical Concerns:**

["NO or VERY MINOR ethics concerns only"]

**Final Justification:**

I hope the authors will be able to include a discussion of the OpenMask3D/Segment3D related work.  For the rest, the issue I had were mostly addressed.  Maybe one small remaining issue, the authors (and many others) talk about training a Gaussian Splat representation, which is nonsense, they are fitting a Gaussian Splat representation.

**Limitations:**

Compared to fully open vocabulary approaches to query scenes, here the method is limited to only return one of the previously reconstructed objects at one of the 3 scales.

**Paper Formatting Concerns:**

Table 1, no time units are given.

**Quality:**

3

**Strengths And Weaknesses:**

3D scene understanding is still a challenging and relevant problem,

but highly related prior work is missing and the heuristics are somewhat limiting (3 scales, how are those determined?), strongly relies on SAM2, how multi-view consistency is achieved is not well explained, seems could easily fail.

The method limits its prior work mostly to Gaussian Splatting-based works, while there is a lot of highly related work with alternative representations that is ignored, OpenScene, OpenMask3D, Segment3D.  In particular the two latter ones are addressing the same problem, but use a pointcloud/mesh representations.

---

> ### Author Rebuttal · Authors · 2025-07-31
>
> Thank you for your review, we sincerely appreciate your comments and would like to clarify the following points.
> # Robustness and Strong Reliance on SAM2
>
> While we use SAM2 to obtain the initial tracking results, our proposed robust tracking module is designed to handle potential failure cases of SAM2. Our pipeline is evaluated across four datasets (e.g., LERF-OVS, 3DOVS, Neu3D, and HyperNeRF), ensuring its robustness. Additionally, we conduct further experiments on the more challenging MipNeRF360 dataset, including scene with hundreds of cluttered objects (e.g., *Counter* and *Bonsai*) to showcase the reliability and effectiveness of our method, the results are provided below:
>
> | Method        | Counter mIoU | Time | Bonsai mIoU | Time | Kitchen mIoU | Time | Garden mIoU | Time | Room mIoU | Time | Avg mIoU | Avg Time |
> |---------------|--------------|------|-------------|------|---------------|------|--------------|------|------------|------|----------|-----------|
> | Ours          | **55.32**    |**36.0**|**63.80**    |**45.0**|**68.59**      |**48.0**|**44.09**      |**86.0**| 42.90      | 44.0 |**54.94** |**51.80**   |
> | OpenGaussian  | 49.57        | 48.0 | 39.08       | 58.0 | 31.22         | 60.0 | 15.10        | 130.0| 41.88      |**42.0**| 35.37    | 67.60     |
> | G-Grouping    | 41.76        | 50.0 | 56.36       | 62.0 | 58.15         | 71.0 | 36.25        | 128.0| 30.84      | 63.0 | 44.67    | 74.80     |
> | LEGaussian    | 54.98        | 50.0 | 62.46       | 52.0 | 66.77         | 60.0 | 40.12        | 103.0|**45.58**    | 61.0 | 53.98    | 65.20     |
>
>
> # Multiview Consistency
>
> We are not entirely sure what the reviewer means by “multiview consistency.” In our pipeline, mask consistency is maintained by SAM2 along with our robust tracking module. Additionally, our proposed object-centric reconstruction ensures that the 3D object geometry is consistent across multiple views, and the CLIP embedding association enforces embedding consistency for the same object across different viewpoints.
>
> # Three scales of SAM
>
> Sorry for the confusion. The 3 scales are semantic levels pre-defined in the Segment Anything (SAM) \[1\] as “whole”, “part” and “subpart”. These levels are used to capture varying degrees of semantic granularity within an object. Please kindly refer to the SAM paper for more details. As these scales are semantic-wise, they are not related to metric (geometric) scales.
>
> \[1\] Kirillov, Alexander, et al. "Segment anything." Proceedings of the IEEE/CVF international conference on computer vision. 2023\.
>
> # Scenes with camera moving close-by and far-away
>
> Thank you for the question. All scenes in the LERF-OVS dataset, on which we conducted our experiments, feature camera trajectories that observe objects both from close-up and from a distance, and our approach performs robustly under these conditions. Please check the LERF \[2\] paper and website for the visualization of their dataset.
>
> \[2\] Kerr J, Kim C M, Goldberg K, et al. Lerf: Language embedded radiance fields\[C\]//Proceedings of the IEEE/CVF international conference on computer vision. 2023: 19729-19739.
>
> # Missing comparison to point cloud-based related works
>
> Thank you for mentioning these related works. However, OpenScene, OpenMask3D, and Segment3D operate under a different problem setting compared to ours. All of these methods assume access to existing 3D geometry (e.g., point clouds) as input, whereas our method takes only multi-view 2D images as input and performs open-vocabulary segmentation after reconstructing the scene geometry.
> As a result, the segmentation performance in our method and the baseline ones inherently influenced by the quality of the reconstruction, introducing a distinct set of challenges not present in the above-mentioned works. Therefore, a direct comparison would not be fair or meaningful, as the input assumptions and task formulations differ significantly. We will go through these works and consider discussing them in the related work section.
>
> # The method is not fully open-vocabulary
>
> Though our method can only return the reconstructed objects at 3 different semantic levels, namely, “whole objects”, “parts”, and “sub-parts”, these levels are designed to **comprehensively cover the semantic granularity** of most objects and their components in the scene.
> As a result, arbitrary text queries can still be matched to the most relevant reconstructed object or part through CLIP-based embedding comparison. While the system does not generate new segmentations on-the-fly, it remains **practically open-vocabulary** in that it supports **flexible and diverse text queries** without being constrained to a fixed category set.

---

> > ### Comment · Reviewer_NzNg · 2025-08-06
> > **response to rebuttal**
> >
> > Dear authors,
> >
> > Thank you for clarifying a few issues, in particular the multi-view consistency and multiple SAM scales that are being used, as well as the fact that the datasets contain object observations at multiple scales.
> >
> > My impression is still that the object tracking and segmentation pipeline is somewhat heuristic and as also mentioned by another reviewer handling dynamic scenes is a bit orthogonal to the open-vocabulary segmentation problem.
> >
> > I also still think that in terms of its open-vocabulary segmentation methodological contribution it is very related to OpenMask3D with Segment3D (as proposed in Segment3D) as the latter extends SAM to 3D and the former associates an ensemble of CLIP descriptors based on object crops while taking object visibility into account.  While I don't think the reconstruction being part of the method or not is an important difference, I agree that the method proposed in this paper goes further as it can potentially generate more complete views of parts as illustrated in Fig 5 and handling dynamic scenes is also a nice plus.
> >
> > I agree that the object, part and subpart scales probably provide a good general set of segments to enable sufficient general open-vocabulary queries.  While proper experimental comparison might be hard given the different datasets, I would still like a discussion related to some of the challenges of using SAM for 3D segmentation as observed in Segment3D.

---

> > > ### Author Response · Authors · 2025-08-07
> > >
> > > We thank the reviewer for his/her valuable feedback and appreciate the opportunity to further clarify the following points.
> > >
> > > ## Object tracking
> > >
> > > Thank you for raising this point. While we acknowledge that our robust object tracking pipeline involves heuristic components (e.g., new object detection, lost track handling), we would like to emphasize that these design choices are intentional and practical, and the robustness of the proposed tracking pipeline have been extensively evaluated across five diverse datasets: four in the main paper (LERF-OVS, 3DOVS, Neu3D, and HyperNeRF) and one additional benchmark in the rebuttal (MipNeRF360). These evaluations demonstrate that our approach remains effective across a wide range of scene types and motion dynamics, reinforcing the practicality of our design.
> > >
> > > ## Dynamic scene handling
> > >
> > > We respectfully disagree with the notion that handling dynamic scenes is orthogonal to open-vocabulary segmentation. In real-world applications, such as robotics, autonomous driving, and AR/VR, scenes are inherently dynamic, with objects continuously entering or changing positions. In such settings, the ability to perform open-vocabulary segmentation in dynamic scenes is essential.
> > >
> > > ## Segment-then-Splat, OpenMask3D and Segment3D
> > >
> > > Thank you for initiating this discussion on 3D segmentation. I would like to first clarify the difference between Segment-then-Splat and OpenMask3D, Segment3D, although they are all focusing on open-vocabulary segmentation.
> > >
> > > OpenMask3D and Segment3D operate under a paradigm where 3D geometry is first obtained, and segmentation is applied afterward.
> > >
> > > In contrast, our Segment-then-Splat framework takes a different approach: we first segment the 3D space into pre-allocated object slots (provided by multiview mask supervision), and then allow the pre-grouped Gaussians to learn to reconstruct each object within its assigned slot. Once the scene is reconstructed, they have already been segmented.
> > >
> > > This can be considered a “chicken-or-the-egg” problem, whether to first reconstruct and then segment, or to segment and then reconstruct. Our approach shows that early segmentation can guide the reconstruction process, resulting in an instance level reconstruction that makes open-vocabulary segmentation easier.
> > >
> > > In summary, our method reconstructs an already segmented 3D scene, while OpenMask3D and Segment3D segment an already reconstructed scene.
> > >
> > > ## Using SAM for 3D segmentation.
> > >
> > > In OpenMask3D and Segment3D, the goal is to find correspondence between point clouds and pixel-based masks generated from SAM, which can be challenging.
> > > In contrast, our method provides this correspondence from the very beginning (before the scene exists). We only require multi-view masks of a given object from SAM2, and can randomly\* sample a set of 3D Gaussians that correspond to this object. During the reconstruction process, this Gaussian-to-object correspondence is explicitly maintained and the group of Gaussians will learn to represent the object. As a result, once reconstruction is complete, the 3D segmentation is inherently available, eliminating the need to resolve mask-to-point correspondences.
> > >
> > > \* Here we use “randomly sample” for better understanding, in our implementation, only objects with no Colmap points are assigned with randomly sampled Gaussians.

---

### Official Review · Reviewer_Yvnk · 2025-06-30

**Clarity:** 3
**Significance:** 3
**Originality:** 3
**Rating:** 4
**Confidence:** 5

**Summary:**

The paper presents Segment then Splat, a 3D-aware open-vocabulary segmentation approach for both static and dynamic scenes based on Gaussian Splatting. Different with traditional splat then segment pipelines used in this task, the proposed method reverses the process by first initializing each object with a specific set of Gaussians. Extensive experiments show state-of-the-art performance across diverse static and dynamic scenes in 3D open-vocabulary segmentation.

**Questions:**

Please refer to the weaknesses.

**Ethical Concerns:**

["NO or VERY MINOR ethics concerns only"]

**Final Justification:**

Thanks for the clarification. The rebuttal addresses most of my concerns. I would suggest the authors review their submission carefully to avoid any ambiguity. Despite this, I appreciate the effort the authors made. Based on these aspects, I will maintain my initial score.

**Limitations:**

yes

**Quality:**

3

**Strengths And Weaknesses:**

Strengths:
+ The writing is easy to follow.
+ The proposed method is reasonable and practical.
+ The proposed method outperforms other baselines in 3D-based segmentation

Weaknesses:
- Since the proposed method is based on 3D segmentation, it is also capable of being evaluated based on the rendered 2D images and then evaluated on 2D segmentation masks. However, Table 1 only includes the 3D-based segmentation results for the proposed method. Including these aspects of experiments can be better for the reader to evaluate the proposed method when compared to other baseline methods, as a common setting in this research area.
- Experiments on Mip-NeRF360. In addition, OpenGaussian offers a new annotated Mip-NeRF360 for 3D open-vocabulary segmentation. The authors should conduct their experiments on this dataset since it contains more views per scene, so that other 2D methods can be better evaluated regardless of the reconstruction tools they used.
- Design. Since the proposed methods are based on per-object segmentation and then conduct reconstruction on it, the inherent limitation of the proposed method is to deal with the text query referring to multiple objects. However, I do not see any discussion or experimental results on this. Offering this kind of information can provide a more comprehensive understanding of these categories of “segment-then-splat” approaches.

---

> ### Author Rebuttal · Authors · 2025-07-31
>
> We thank the reviewer for recognizing the reasonableness, practicality and performance of our proposed method. Your support is greatly appreciated and we would like to take this opportunity to clarify several points.
>
> # \[W1\] Evaluation on 2D Segmentation Mask
>
> Thank you for pointing this out, and we apologize for the confusion. The evaluation results presented in Tab.1 are computed based on **2D segmentation masks**, in order to align with the evaluation protocols used by all baseline methods.
> For the 3D part in Tab.1, we modify 2D-based baselines (e.g., LEGaussian and LangSplat) to perform segmentation directly on **3D Gaussians** instead of pixels. However, to maintain a fair and consistent comparison, **all methods are ultimately evaluated on their rendered 2D segmentation maps**. This ensures comparability across different approaches while preserving the integrity of 3D-aware segmentation modeling.
>
> # \[W2\] Experiments on Mipnerf360
>
> Thank you for your suggestion. We have conducted additional experiments on the Mipnerf360 dataset. And the results are shown in the table below, where our method performs well on this more challenging dataset and surpasses all baseline methods:
>
> | Method        | Counter mIoU | Time | Bonsai mIoU | Time | Kitchen mIoU | Time | Garden mIoU | Time | Room mIoU | Time | Avg mIoU | Avg Time |
> |---------------|--------------|------|-------------|------|---------------|------|--------------|------|------------|------|----------|-----------|
> | Ours          | **55.32**    |**36.0**|**63.80**    |**45.0**|**68.59**      |**48.0**|**44.09**      |**86.0**| 42.90      | 44.0 |**54.94** |**51.80**   |
> | OpenGaussian  | 49.57        | 48.0 | 39.08       | 58.0 | 31.22         | 60.0 | 15.10        | 130.0| 41.88      |**42.0**| 35.37    | 67.60     |
> | G-Grouping    | 41.76        | 50.0 | 56.36       | 62.0 | 58.15         | 71.0 | 36.25        | 128.0| 30.84      | 63.0 | 44.67    | 74.80     |
> | LEGaussian    | 54.98        | 50.0 | 62.46       | 52.0 | 66.77         | 60.0 | 40.12        | 103.0|**45.58**    | 61.0 | 53.98    | 65.20     |
>
>
> # \[W3\] Text query referring to multiple objects
>
> Yes, this is indeed a limitation of our current method. Since object embeddings are derived solely from objects themselves, our model does not explicitly capture spatial or relational context between multiple objects. However, this limitation is shared by all baseline methods \[1\]\[2\]\[3\]\[4\]\[5\], as their embeddings are constructed in a similar object-isolated manner.
> Incorporating a global scene-level representation, such as a scene graph that models relationships between objects (e.g., proximity, co-occurrence, or functional interaction), could potentially address this issue. We consider this a promising direction for future work and appreciate the reviewer for highlighting this important point.
>
> \[1\] Shi J C, Wang M, Duan H B, et al. Language embedded 3d gaussians for open-vocabulary scene understanding\[C\]//Proceedings of the IEEE/CVF Conference on Computer Vision and Pattern Recognition. 2024: 5333-5343.
> \[2\] Qin M, Li W, Zhou J, et al. Langsplat: 3d language gaussian splatting\[C\]//Proceedings of the IEEE/CVF Conference on Computer Vision and Pattern Recognition. 2024: 20051-20060.
> \[3\] Ye M, Danelljan M, Yu F, et al. Gaussian grouping: Segment and edit anything in 3d scenes\[C\]//European conference on computer vision. Cham: Springer Nature Switzerland, 2024: 162-179.
> \[4\] Wu Y, Meng J, Li H, et al. Opengaussian: Towards point-level 3d gaussian-based open vocabulary understanding\[J\]. Advances in Neural Information Processing Systems, 2024, 37: 19114-19138.
> \[5\] Labe I, Issachar N, Lang I, et al. Dgd: Dynamic 3d gaussians distillation\[C\]//European Conference on Computer Vision. Cham: Springer Nature Switzerland, 2024: 361-378.

---

> > ### Comment · Reviewer_Yvnk · 2025-08-06
> > **W3**
> >
> > Thanks for the rebuttal. The rebuttal addresses most of my concerns in W1 and W2. However, for W3, I do not agree that existing methods will be severely limited in multiple-object queries since they do not explicitly model the per-object segmentation as an initial step of the proposed method. I would like to recommend that the authors make their justifications carefully first. Second, it is uncertain how the proposed method performs when the rendered scene has multiple objects with the same semantic category. Besides, considering some common queries like the abstract query `electricity` used in LERF's teaser figure, if such a concept covers multiple objects in a single view, the proposed method may not be able to handle such a case. This makes me feel that the proposed method is somehow oriented to a limited experimental setting but not really for real-world cases, which is essential for open-vocabulary applications. Discussions are welcome.

---

> > > ### Author Response · Authors · 2025-08-06
> > >
> > > We thank the reviewer for his/her response and appreciate the opportunity to further clarify our method. We apologize for the earlier misunderstanding regarding what was meant by “text query referring to multiple objects.” Below, we provide detailed clarifications addressing each of the concerns raised.
> > >
> > > ## 1. Text query that refers to multiple objects with the same semantic category
> > >
> > > We acknowledge the reviewer’s point and agree that supporting multi-object queries is essential for open-vocabulary segmentation. While our current inference strategy is designed to retrieve one object with the highest semantic similarity to the text embedding, this strategy can be easily extended. We can change this strategy by setting a threshold to the similarity between Gaussian embeddings and the input text embedding, and retrieve all Gaussian groups whose embeddings exceed the threshold. This way, multiple relevant objects can be retrieved. A similar thresholding mechanism is also adopted by methods like LEGaussian and LangSplat, where per-pixel feature embeddings are compared against the query embedding, and pixels with similarity above a certain threshold are selected to form a segmentation mask.
> > >
> > > ## 2. Abstract query
> > >
> > > Our method is capable of handling abstract queries such as “electricity.” The retrieval is based on CLIP embedding similarity. While abstract concepts like “electricity” may not correspond to a single, clearly defined object, semantically related objects such as “power outlets,” “power strips,” or “light switches” typically have embeddings with high cosine similarity (e.g., 0.85 for “power outlet”, 0.79 for “power strips” and 0.80 for “light switches”) to the abstract query. Furthermore, if multiple related objects related to the abstract query are present in the scene, they can all be selected using the thresholding mechanism mentioned above.
> > >
> > > ## 3. Limitation of our methods and all the baselines
> > >
> > > The limitation of our method and all the baseline methods we talked about in the rebuttal is “text query that contains relationship from multiple objects”, for example, “a sheep sitting on the chair in front of the table”.  In our method (as well as in LangSplat, OpenGaussian, and GaussianGrouping), the textual embedding is derived from masked regions corresponding to individual objects, and thus lacks relational or contextual understanding. Similarly, LEGaussian relies on local image patch features, which also doesn’t model object-to-object relationships.
> > >
> > > We believe this is an important direction for future work. Incorporating scene-level relational reasoning, possibly through the use of scene graphs, may help bridge this gap and enable more expressive open-vocabulary querying capabilities.

---

> > > > ### Comment · Reviewer_Yvnk · 2025-08-07
> > > >
> > > > Thanks for the feedback. While I agree that the proposed method can retrieve multiple objects by applying a threshold, this post-processing approach may degrade performance in single-object retrieval scenarios and may still not be suitable for real-world applications. Nevertheless, I appreciate the authors' effort and thoughtful comments. I have finalized my rating and encourage the authors to revise their submission based on my comments.

---

> > > > > ### Author Response · Authors · 2025-08-07
> > > > >
> > > > > Thank you for your response and valuable advice. However, we respectfully disagree with the notion that our proposed method is unsuitable for real-world applications.
> > > > >
> > > > > The thresholding post-processing strategy we employ is a common practice in many state-of-the-art methods, including LERF [1], LEGaussian [2], LangSplat [3], and GaussianGrouping [4]. Furthermore, several works have successfully applied LERF in real-world settings, such as robotic grasping [5] and inventory monitoring [6], demonstrating the practical effectiveness of this approach.
> > > > >
> > > > > To further support our claim, we modified our inference strategy to incorporate thresholding and conducted an additional experiment on the LERF-OVS dataset:
> > > > >
> > > > > | Method, mIoU         | waldo_kitchen |   ramen   |  teatime  | figurines | average   |
> > > > > |----------------------|:-------------:|:---------:|:---------:|:---------:|-----------|
> > > > > |       LangSplat      |     36.57     |   42.62   |   60.29   |   45.99   |   46.37   |
> > > > > |      LEGaussians     |     17.80     |   11.64   |   44.72   |    1.03   |   18.79   |
> > > > > |      G-Grouping      |     11.60     |   31.13   |   46.83   |   28.79   |   29.59   |
> > > > > |     OpenGaussian     |     32.37     |   22.20   |   60.82   | **54.31** |   42.43   |
> > > > > | Ours (thresholding)  |   **38.04**   | **49.41** | **61.04** |   46.98   | **48.86** |
> > > > > | Ours (single object) |     40.71     |   54.38   |   63.47   |   49.83   |   52.10   |
> > > > >
> > > > >
> > > > > While the single-object retrieval performance shows some degradation when applying thresholding, it still surpasses all baseline methods. For this experiment, the threshold was set at 85% of the highest object similarity found in the scene for each query.
> > > > >
> > > > > Thank you again for your valuable feedback. We hope this additional evidence will help reconsider the real-world applicability of our method.
> > > > >
> > > > >
> > > > > [1] Kerr, Justin, et al. "Lerf: Language embedded radiance fields." Proceedings of the IEEE/CVF international conference on computer vision. 2023.
> > > > > [2] Shi, Jin-Chuan, et al. "Language embedded 3d gaussians for open-vocabulary scene understanding." Proceedings of the IEEE/CVF Conference on Computer Vision and Pattern Recognition. 2024.
> > > > > [3] Qin, Minghan, et al. "Langsplat: 3d language gaussian splatting." Proceedings of the IEEE/CVF Conference on Computer Vision and Pattern Recognition. 2024.
> > > > > [4] Ye, Mingqiao, et al. "Gaussian grouping: Segment and edit anything in 3d scenes." European conference on computer vision. Cham: Springer Nature Switzerland, 2024.
> > > > > [5] Rashid, Adam, et al. "Language embedded radiance fields for zero-shot task-oriented grasping." 7th Annual Conference on Robot Learning. 2023.
> > > > > [6] Rashid, Adam, et al. "Lifelong lerf: Local 3d semantic inventory monitoring using fogros2." 2024 IEEE International Conference on Robotics and Automation (ICRA). IEEE, 2024.

---

### Official Review · Reviewer_TP1s · 2025-07-01

**Clarity:** 3
**Significance:** 3
**Originality:** 2
**Rating:** 4
**Confidence:** 3

**Summary:**

The paper proposes a 3D open-vocabulary segmentation method called "Segment then Splat", which reverses the traditional "Splat-then-Segment" method. Instead of conducting segmentation after scene reconstruction, the method segments objects at the 3D Gaussian initialization stage, ensuring object-specific Gaussian sets before reconstruction. After reconstruction, semantic embeddings from CLIP are assigned to each object to enable open-vocabulary querying.

**Questions:**

1. Scalability:

How does the method perform on large-scale scenes with many objects? Does the computational cost scale linearly or non-linearly with the number of objects?

2. Robustness of Object Tracking:

In cases where SAM2 fails to track objects accurately (e.g., cluttered scenes or objects with similar appearances), how does this affect the segmentation and reconstruction performances?

3. Dynamic Scene Complexity:

How does the method handle highly dynamic scenes with significant object deformation or occlusion? Are there some failure cases?

4. Post-Processing Dependence:

The partial mask filtering strategy appears to be a post-hoc correction for mask inaccuracies. How significant is its impact on the final results, and does this suggest limitations in the present pipeline?

**Ethical Concerns:**

["NO or VERY MINOR ethics concerns only"]

**Final Justification:**

I have carefully read the rebuttal. Thank the authors for their efforts and thorough analysis. Most of my concerns have been addressed. However, I still think the novelty of this work is somewhat limited. As they pointed out, the key idea of this work is to "reverse the long-established 'Splat then Segment' approach." From my perspective, this represents only a minor contribution to novelty, as it appears to be a relatively straightforward modification. Nonetheless, I appreciate the experiments they conducted to address the concerns regarding efficiency.

In addition, the responses to Q2, Q3, and Q4 have effectively resolved my concerns. Finally, I would suggest including more detailed discussions on Open-Vocabulary 3D detection in your revision. Doing so will not diminish the contribution but will instead benefit the community in terms of both OV-3DSeg and OV-3DDet.

Overall, I will raise my score to 4.

**Limitations:**

Yes, the authors have discussed the limitations on page 9.

**Quality:**

3

**Strengths And Weaknesses:**

**Strengths**

1. Reversal of the Traditional Pipeline:

The proposed "Segment then Splat" pipeline introduces a straightforward modification to the traditional "Splat-then-Segment" approach, addressing issues like Gaussian-object misalignment in dynamic scenes.

2. Dynamic Scene Support:

By enforcing object-Gaussian correspondence and using a strong object tracking module, the method gains the ability to handle dynamic scenes better than other methods.

3. Comprehensive Evaluation:

Experiments on both static and dynamic datasets provide sufficient evidence of the method’s effectiveness compared to other methods.

4. Qualitative Results:

The paper demonstrates sharper segmentation boundaries and better object retrieval performance, particularly in cases where previous methods struggled with multi-object scenes or dynamic scenes.

**Weaknesses**

1. Limited Novelty:

The "Segment then Splat" pipeline is conceptually simple, consisting of two straightforward steps:
- Perform 3D segmentation on point clouds before reconstruction.
- Use CLIP embeddings for semantic matching.
This method lacks a unique technique design, as it essentially combines existing techniques (SfM-based segmentation + CLIP-based matching) in a straightforward manner.

2. Pipeline Complexity:

The method requires per-object 3DGS reconstruction, which makes the pipeline computationally expensive and potentially impractical for large-scale or complex scenes with many objects.

3. Scalability Concerns:

The computational efficiency of the method for scenes with hundreds or thousands of objects is unclear, as the need for object-specific Gaussian reconstruction could introduce lots of overhead.

4. Lacking Discussions about the OV 3D Detection settings

Considering the research contexts are very related, the submission should also discuss the relation with methods of OV-3DDet settings [1, 2, 3]. I'm curious about their differences and whether this method has the potential for OV-3DDet applications


[1] Pengkun Jiao, Na Zhao, Jingjing Chen, and Yu-Gang Jiang. Unlocking Textual and Visual Wisdom: Open-Vocabulary 3D Object Detection Enhanced by Comprehensive Guidance from Text and Image. In ECCV, 2025.

[2] Yang Cao, Yihan Zeng, Hang Xu, and Dan Xu. CoDA: Collaborative Novel Box Discovery and Cross-Modal Alignment for Open-Vocabulary 3D Object Detection. In NeurIPS 2023.

[3] Yuheng Lu, Chenfeng Xu, Xiaobao Wei, Xiaodong Xie, Masayoshi Tomizuka, Kurt Keutzer, and Shanghang Zhang. Open-Vocabulary Point-Cloud Object Detection without 3D Annotation. In CVPR 2023

---

> ### Author Rebuttal · Authors · 2025-07-31
>
> We thank the reviewer for the thoughtful feedback and for recognizing our proposed “Segment-then-Splat” approach as a novel reversal of the traditional pipeline, the comprehensiveness of our evaluation, and our method’s ability to support dynamic scenes. We would like to take this opportunity to clarify the following points.
>
> # \[W1\] Novelty
>
> We appreciate the reviewer’s perspective. While our “Segment then Splat” pipeline may appear conceptually simple, we would like to argue in the following aspects:
>
> 1) We reverse the long established “Splat then Segment” approach in Gaussian Splatting based open-vocabulary segmentation tasks.
> 2) Based on (1) we establish an object-centric reconstruction approach, achieving a better object boundary localization as shown in Fig.4 compared to baseline “Splat then Segment” methods. Unlike prior methods, our approach does not require training an additional feature field, making it more efficient and easier to optimize.
> 3) Based on (1), we eliminate the Gaussian-object misalignment issue in dynamic modeling, thus can be directly applied to dynamic scenes without additional modification to form a unified 3D open-vocabulary segmentation approach.
>
> # \[W2, W3, Q1\] Efficiency, Scalability and Performance on Larger Scenes
>
> Thank you for pointing this out. In the current implementation we proposed a random sampling strategy to compensate for the overhead introduced by object-specific supervision during training. To evaluate the scalability of our method, we conducted additional experiments on the MipNeRF360 \[2\] dataset, which includes highly cluttered scenes with hundreds of objects (e.g., Bonsai and Counter). The performance and efficiency metrics are summarized in the following table:
>
> | Method        | Counter mIoU | Time | Bonsai mIoU | Time | Kitchen mIoU | Time | Garden mIoU | Time | Room mIoU | Time | Avg mIoU | Avg Time |
> |---------------|--------------|------|-------------|------|---------------|------|--------------|------|------------|------|----------|-----------|
> | Ours          | **55.32**    |**36.0**|**63.80**    |**45.0**|**68.59**      |**48.0**|**44.09**      |**86.0**| 42.90      | 44.0 |**54.94** |**51.80**   |
> | OpenGaussian  | 49.57        | 48.0 | 39.08       | 58.0 | 31.22         | 60.0 | 15.10        | 130.0| 41.88      |**42.0**| 35.37    | 67.60     |
> | G-Grouping    | 41.76        | 50.0 | 56.36       | 62.0 | 58.15         | 71.0 | 36.25        | 128.0| 30.84      | 63.0 | 44.67    | 74.80     |
> | LEGaussian    | 54.98        | 50.0 | 62.46       | 52.0 | 66.77         | 60.0 | 40.12        | 103.0|**45.58**    | 61.0 | 53.98    | 65.20     |
>
> As shown, our method outperforms baseline approaches without any modification to the training pipeline, while maintaining superior efficiency. This suggests that our random sampling strategy effectively balances supervision and training cost, even in more complex scenes.
>
> However, such overhead on larger scenes can still increase linearly if we need either more objects to be sampled per iteration or extend the training iteration when facing extreme cases.
> To further improve the training efficiency, the rasterization backend can be modified to support batch rendering on the objects (e.g. one pass rasterization for all the objects). Notably, the GSplat \[1\] backend already supports batch rasterization for a single set of Gaussians. By modifying the rasterizer to add Gaussian masks during rasterization, we can render different objects in batch, thus enabling a more efficient and scalable training process.  Although we have not implemented this yet, we consider it a promising future refinement, and we believe the efficiency issue can be fully resolved through such engineering strategies.
>
> \[1\] Ye V, Li R, Kerr J, et al. gsplat: An open-source library for Gaussian splatting\[J\]. Journal of Machine Learning Research, 2025, 26(34): 1-17.
>
> # \[Q2\] Robustness of Object Tracking
>
> SAM2 itself may easily fail to track objects accurately in complex scenarios (e.g., *Figurines* in Tab.2(c)), due to issues like lost tracking, duplicate tracking etc.. And that’s the reason why we designed a robust tracking module to handle these issues, ensuring reliable supervision for the following object-specific reconstruction. And we also conduct further experiment on MipNeRF360 dataset to demonstrate the reliability of our tracking module on complex scenes with cluttered objects, please refer to **the table in the above section**.
>
> # \[Q3\] Highly Dynamic and Occluded Scenes
>
> The ability to handle significant object deformation primarily depends on the underlying 4D Gaussian Splatting (4DGS) method used. Our approach is compatible with any 4DGS pipeline, thus, the capacity to model complex dynamics is inherently limited by the dynamic modeling capabilities of the chosen 4DGS method.
> For highly occluded scenarios, the *Figurines* in the LERF-OVS dataset and *Kitchen* in MipNeRF360 dataset can serve as representative examples. In these challenging settings, our proposed pipeline and tracking method demonstrate better performance over baseline methods.
>
> # \[Q4\] Partial Mask Filtering
>
> Sorry for the confusion. Partial mask filtering is not a correction for mask inaccuracies. The mask provided by our tracking module is accurate enough. The motivation for incorporating this strategy is that the rendered object Gaussians may reveal the full object structure, even in views where the object is heavily occluded, as it can learn from other views. A demonstration is shown in Fig. 5 of the main paper. In such cases, direct supervision using the SAM mask from the occluded view would impose an incorrect constraint, potentially distorting the learned object structure. Partial mask filtering helps avoid this by preventing such misleading supervision signals. The ablation study for partial mask filtering **has already been done** in main paper Tab. 2(b). In highly occluded scenes such as *Ramen* and Waldo-Kitchen, partial mask filtering significantly improves both segmentation and reconstruction performance.
>
> # \[W4\] Discussion on Open-Vocabulary 3D Detection
>
> Thank you for pointing this out. Open-vocabulary 3D detection and open-vocabulary 3D segmentation are indeed closely related tasks. The key distinction lies in the level of spatial precision required: 3D segmentation aims to recover accurate **object boundaries**, whereas 3D detection only requires **coarse localization via bounding boxes**. Our approach can naturally be extended to open-vocabulary 3D detection by simply computing a 3D bounding box that encloses the extracted object-specific Gaussians.

---

> > ### Comment · Reviewer_TP1s · 2025-08-04
> > **Comments for the authors**
> >
> > Dear authors, I have carefully read your comments. Thank you for your efforts and thorough analysis. Most of my concerns have been addressed. However, I still think the novelty of this work is somewhat limited. As you pointed out, the key idea of this work is to "reverse the long-established 'Splat then Segment' approach." From my perspective, this represents only a minor contribution to novelty, as it appears to be a relatively straightforward modification. Nonetheless, I appreciate the experiments you conducted to address the concerns regarding efficiency.
> >
> > In addition, your responses to Q2, Q3, and Q4 have effectively resolved my concerns. Finally, I would suggest including more detailed discussions on Open-Vocabulary 3D detection in your revision. Doing so will not diminish your contribution but will instead benefit the community in terms of both OV-3DSeg and OV-3DDet.
> >
> > Overall, I will raise my score to 4.

---

> > > ### Author Response · Authors · 2025-08-04
> > >
> > > Thank you very much for your careful consideration and valuable feedback. We are glad that our response has addressed your concerns. Nevertheless, we would like to take this opportunity to further clarify and emphasize our contribution.
> > >
> > > All prior methods (e.g., OpenGaussian, LangSplat, Gaussian Grouping, etc.) follow a “Splat-then-Segment” paradigm. Taking OpenGaussian as an example, it first reconstructs the scene using 3D Gaussian Splatting while simultaneously learning a feature field via contrastive loss. It then clusters the Gaussians based on the learned features, and finally assigns CLIP embeddings to each Gaussian to enable open-vocabulary segmentation.
> > >
> > > “Splat-then-Segment” will cause several issues.
> > >
> > > 1. Geometric and Semantic Ambiguities: Without object-level constraints during reconstruction, Gaussians near object boundaries may represent multiple objects and carry mixed semantic information, leading to ambiguity.
> > > 2. Object-Gaussian Misalignment in Dynamic Scenes: In dynamic scenes, a single Gaussian may represent different objects across time, requiring impractical time-varying embeddings for open-vocabulary segmentation. Thus, direct adaptation of “Splat-then-Segment” methods to dynamic scenes is fundamentally limited.
> > >
> > > Our core motivation is to achieve superior open-vocabulary segmentation performance by **eliminating the geometric and semantic ambiguities** associated with each Gaussian. To address this, we reverse the traditional “Splat-then-Segment” approach and establish a novel **Segment-then-Splat** approach, here we highlight the goal for each step using **bold**, while the proposed components are highlighted using ***bold & italic***:
> > >
> > > 1. First, we segment Gaussians into distinct groups representing individual objects through our ***robust tracking module***, **establishing clear object-Gaussian correspondence**.
> > > 2. Second, during object-centric reconstruction, we enforce the object-Gaussian correspondence by providing *per-object/part supervision*. This **eliminates geometric ambiguity** since each Gaussian exclusively represents a single object or part. Additionally, we introduce an ***optimization strategy across multiple granularities*** to preserve structural completeness. A ***partial mask filtering strategy*** is also proposed to align the optimization goal (entire 3D object structure) with the supervision signal (2D masks), further enhancing reconstruction quality.
> > > 3. Finally, we assign each Gaussian group a consistent language embedding derived from multiview object images through ***CLIP embedding association***, ensuring semantic consistency and completely **eliminating semantic ambiguity**.
> > > 4. Additionally, thanks to the object-Gaussian correspondence we establish, our method can be easily **adapted to Dynamic scenes** without further modification.
> > >
> > > We also genuinely appreciate your insightful suggestion to expand the discussion on Open-Vocabulary 3D Detection (OV-3DDet). We will certainly incorporate a more detailed exploration of OV-3DDet in our revised manuscript, as we agree this will enrich the paper and provide further utility to researchers in both OV-3DSeg and OV-3DDet domains.
> > >
> > > Thank you once again for your constructive review, which has significantly improved our manuscript.

---

### Official Review · Reviewer_5uyD · 2025-07-01

**Clarity:** 3
**Significance:** 3
**Originality:** 3
**Rating:** 4
**Confidence:** 4

**Summary:**

The paper introduces "Segment then Splat," a novel 3D open-vocabulary segmentation method using Gaussian Splatting that segments objects before reconstruction. This reverses the traditional "splat then segment" approach, maintaining precise object-Gaussian correspondence for improved 3D geometry and segmentation accuracy. It supports both static and dynamic scenes by preventing Gaussian-object misalignment and includes a robust object tracking module for spatial-temporal consistency. By associating CLIP embeddings with object-specific Gaussians, it enables efficient open-vocabulary querying without extra language fields. Experiments show state-of-the-art performance in accuracy and efficiency across diverse datasets

**Questions:**

1. I noticed that the training process adopts a three-stage optimization strategy, progressively encouraging the model to focus on objects of increasing scale, from small parts to larger wholes. However, the paper seems to lack a detailed experimental analysis that specifically evaluates the model’s performance on different object hierarchies or granularities, particularly for relatively smaller subparts. I am especially interested in understanding whether the model can accurately identify and segment these fine-grained, smaller objects (subparts) with precision comparable to larger objects. Such an analysis would provide valuable insights into the robustness and granularity of the proposed segmentation approach.

2. If possible, I would like to hear an explanation about Weaknesses 3 above. Of course, if my understanding is incorrect, please feel free to point it out.

3. During training, the number of supervised objects per iteration is identified as a crucial hyperparameter influencing segmentation performance, as shown in the ablation study in paper. It appears that in none of the static scenes does performance fully saturate with increasing numbers of supervised objects. I wonder if this observation might be due to insufficient supervision over certain part and subpart-level objects, which may dilute the learning signal when only random sampling is employed. If possible, I kindly request the authors to provide some analysis or insight into this aspect.

4. It would be helpful if the authors could include some failure cases or typical scenarios where the proposed method does not perform satisfactorily, in order to better understand its limitations and potential areas for improvement.

**Ethical Concerns:**

["NO or VERY MINOR ethics concerns only"]

**Final Justification:**

Overall, I find the method proposed in this paper to be novel. Although it inevitably involves a lot of engineering and somewhat complicated design, I lean towards a weak accept.

**Limitations:**

Yes

**Quality:**

3

**Strengths And Weaknesses:**

#### Strengths:

1. The paper is clearly written and well-structured, making it easy to follow.

2. The concept of “Segment then Splat” represents a highly innovative and bold departure from the conventional strategy of explicitly modeling semantic fields in open-vocabulary Gaussian approaches. By reversing the traditional “splat then segment” paradigm, this method fundamentally eliminates the reliance on separate language fields for semantic understanding. Moreover, it effectively addresses challenges such as the misalignment between Gaussians and objects, particularly prominent in dynamic scenes. The subsequent experimental results robustly validate the effectiveness of this approach, demonstrating its practical viability and advantages over prior methods.

3. The experimental performance is impressive: across a variety of both static and dynamic datasets, Segment then Splat achieves state-of-the-art results in 3D open-vocabulary segmentation as well as computational efficiency. Extensive qualitative visualizations further illustrate the model’s ability to preserve the integrity of 3D structures while delivering sharp, well-defined object boundaries.

4. Another strength of the proposed framework is its natural compatibility with dynamic scenes. By maintaining a strict one-to-one correspondence between Gaussians and objects, the method effectively prevents Gaussian-object misalignment caused by object motion. This capability allows Segment then Splat to seamlessly unify the treatment of both static and dynamic environments, overcoming a major limitation of existing methods that struggle with time-varying or moving objects.

#### Weaknesses:

1. Although the paper adopts the Segment then Splat approach to avoid explicitly incorporating semantic parameters into Gaussians, this increases complexity during the preprocessing stage. Additionally, at inference time, the model still requires some matching method to assign semantic information to Gaussians. This leads to a more fragmented model pipeline rather than an end-to-end framework.

2. For more complex scenes, compared to explicitly embedding semantic information into Gaussians, this method may be more difficult to handle. This is because the grouping prior relies heavily on high-precision grouping initialization and complex tracking logic, which may be fragile or error-prone in challenging scenarios.

3. If I understand correctly, during experiments comparing with baselines, the ground truth used is at the object-level rather than pixel-level. This might introduce some unfairness to methods optimized at pixel-level such as LangSplat. If my understanding is incorrect, please kindly correct me. Otherwise, providing additional quantitative comparison with pixel-level ground truth would be beneficial. Of course, due to techniques like Partial Mask Filtering used during training, some performance gap is expected and a slight disadvantage could be acceptable.

4. Although the paper includes a section discussing limitations, it does not comprehensively illustrate failure cases or conditions under which the method’s performance degrades. Presenting such results would help readers gain a more complete understanding of the model’s robustness and boundaries.

---

> ### Author Rebuttal · Authors · 2025-07-31
>
> Thank you for your thoughtful and encouraging review. We sincerely appreciate your valuable feedback and your recognition of our method's novelty and strong performance. We are delighted that you highlighted the innovation of the "Segment then Splat" concept and acknowledged its advantages for dynamic scenes. Your support is very motivating.
>
> # \[W1\] More fragmented model pipeline.
>
> Although our method introduces a more complex preprocessing step, it eliminates the need to train an additional feature field. Moreover, the CLIP embedding association is performed during training, not at inference time. At inference, we simply compare the input text embedding with the precomputed embeddings of each Gaussian group to retrieve the most relevant object, ensuring an efficient and lightweight retrieval process.
>
> # \[W2\] Complex scenes may be challenging.
>
> In complex scenes where semantic information is dense and noisy, language field-based methods tend to struggle with entangled representations. In contrast, our approach associates language embedding with object-specific Gaussian groups, leading to more robust and disentangled representation.
>
> The tracking logic has shown reliable performance on LERF-OVS, 3DOVS, Neu3D and HyperNeRF dataset. We conduct additional experiment on MipNeRF360 dataset, which contains scenes with hundreds of cluttered objects (i.e. Counter and Bonsai), to demonstrate the effectiveness and reliability of our approach:
>
> | Method        | Counter mIoU | Time | Bonsai mIoU | Time | Kitchen mIoU | Time | Garden mIoU | Time | Room mIoU | Time | Avg mIoU | Avg Time |
> |---------------|--------------|------|-------------|------|---------------|------|--------------|------|------------|------|----------|-----------|
> | Ours          | **55.32**    |**36.0**|**63.80**    |**45.0**|**68.59**      |**48.0**|**44.09**      |**86.0**| 42.90      | 44.0 |**54.94** |**51.80**   |
> | OpenGaussian  | 49.57        | 48.0 | 39.08       | 58.0 | 31.22         | 60.0 | 15.10        | 130.0| 41.88      |**42.0**| 35.37    | 67.60     |
> | G-Grouping    | 41.76        | 50.0 | 56.36       | 62.0 | 58.15         | 71.0 | 36.25        | 128.0| 30.84      | 63.0 | 44.67    | 74.80     |
> | LEGaussian    | 54.98        | 50.0 | 62.46       | 52.0 | 66.77         | 60.0 | 40.12        | 103.0|**45.58**    | 61.0 | 53.98    | 65.20     |
>
> # \[W3, Q2\] Object-level ground truth is not fair for those pixel-level methods.
>
> Sorry for the confusion. During the experiments the ground truth used is actually **pixel-level**, not object-level. This aligns with the original setting of all the baselines. For 3D comparison with LangSplat and LEGaussian, we modify their inference code to extract text-related Gaussians instead of pixels and render the extracted Gaussians to image plan to evaluate on pixel-level ground truth. Based on this evaluation strategy, the performance of the original baselines will not degrade, thus the comparison is fair for baseline methods. While our approach may suffer from small performance degradation due to the pixel-level ground truth.
>
> # \[Q1\] Performance analysis on different object granularities.
>
> Thank you for pointing this out. We separate the ground truth objects in LERF\_OVS dataset into two categories: 1\) large/whole objects (e.g. ramen, bear, sheep), represented by (L), 2\) small/sub-parts (e.g. onion segment (in the ramen), bear nose, hooves (of the sheep)), represented by (S). And provide the quantitative results in the following table:
>
> | Method         | Ramen (L) | Ramen (S) | Kitchen (L) | Kitchen (S) | Teatime (L) | Teatime (S) | Figurines (L) | Figurines (S) | Avg (L) | Avg (S) |
> |----------------|-----------|-----------|--------------------|--------------------|-------------|-------------|----------------|----------------|---------|---------|
> | Ours           | **47.24** | **59.22** | **51.60**          | **3.68**           | **69.42**   | **53.58**   | 54.61          | 33.11          | **55.72**|**41.06**|
> | OpenGaussian   | 27.85     | 0.16      | 41.89              | 0.00               | 64.67       | 52.82       | **59.96**      | **33.57**      | 48.59   | 27.03   |
> | G-Grouping     | 47.17     | 2.26      | 15.01              | 0.00               | 60.37       | 37.90       | 59.02          | 1.68           | 45.39   | 17.45   |
> | LEGaussian     | 18.69     | 4.69      | 23.00              | 0.00               | 42.38       | 48.73       | 0.90           | 1.13           | 21.24   | 15.16   |
>
>
> # \[Q3\] Supervision may be insufficient due to object random sampling.
>
> Thank you for the excellent point. You are correct that random sampling can dilute the supervision signal. In our settings, we chose to supervise 3 objects to balance the performance and optimization efficiency. As shown in Tab.2(a), while performance does not fully saturate, we observe a clear convergence trend: the improvement from supervising 1 to 3 objects is substantial, while gains from 3 to 5 (and beyond) become marginal. In scenes with a large number of objects (e.g., *Ramen*), achieving full saturation may require supervising more objects or running additional optimization iterations.
>
> # \[W4, Q4\] Limitations and potential improvements.
>
> One limitation is that sometimes the associated embeddings with each Gaussian set cannot be successfully queried by text-input. This is likely due to our current use of a simple averaging strategy when assigning CLIP embeddings. In cases where an object is heavily occluded in certain viewpoints, the embedding from that view may ruin the entire object embedding and result in query failure. This is also a future direction, developing a more robust and adaptive embedding assignment strategy could further enhance retrieval accuracy and overall performance.

---

> > ### Comment · Reviewer_5uyD · 2025-08-06
> >
> > Thank you for your detailed rebuttal, which has addressed most of my concerns. However, I still have one remaining question regarding the inference strategy you propose. As you mentioned in [W3, Q2], "Based on this evaluation strategy, the performance of the original baselines will not degrade." While this seems reasonable in general, I am uncertain whether this holds for methods like LangSplat, which rely heavily on differentiable rendering during training to learn semantics. If, during inference, semantic labels are obtained by filtering and selecting matched Gaussians rather than through the learned differentiable rendering process, wouldn’t this potentially affect the model's performance? I would appreciate further clarification on this.

---

> ### Author Response · Authors · 2025-08-06
>
> Thank you for your response. In Table 1(a), our evaluation strategy aligns with the original settings of the following baselines: LangSplat (2D), LEGaussian (2D), G-Grouping (2D), and OpenGaussian (3D), ensuring their reported performance remains unaffected. However, for LEGaussian (3D) and LangSplat (3D), which rely on differentiable rendering to learn language embeddings for each Gaussian, our evaluation setup may lead to a degradation in performance. Due to the lack of other suitable 3D-based baselines for comparison, we adopt the experimental setting used in OpenGaussian[1] to compare with LEGaussian (3D) and LangSplat (3D). For details, please kindly refer to Section 4.1(2) Baseline in the OpenGaussian paper.
>
> [1] Wu, Yanmin, et al. "Opengaussian: Towards point-level 3d gaussian-based open vocabulary understanding." Advances in Neural Information Processing Systems 37 (2024): 19114-19138.

---

> > ### Comment · Reviewer_5uyD · 2025-08-08
> >
> > Thank you for your reply. Most of my questions have been resolved, and I am also willing to maintain my positive score.

---

### Note · Authors · 2025-08-13

Dear AC and reviewers,

We sincerely thank the reviewers and ACs for their effort and constructive feedback during the discussion.

Below is a summary of the main concerns raised by the reviewers in the initial review and what we clarified.

* **Novelty and contribution.** Our method reverses the traditional “Splat-then-Segment” pipeline into a novel “Segment-then-Splat” approach. By explicitly establishing Gaussian-object correspondence, it resolves geometry and semantic ambiguities during reconstruction, enabling more accurate language embedding associations and improved open-vocabulary segmentation performance.  Additionally, our method eliminates the need to train a feature field, resulting in a more efficient pipeline, and can be directly adapted to dynamic scenes without any further modifications.
* **Robustness and scalability on more complex scenes.** In the main paper, we present quantitative experiments on four diverse datasets, including both static and dynamic scenes. During rebuttal, we evaluate our method on the MipNeRF 360 dataset during rebuttal, which features highly cluttered environments with hundreds of objects, to further demonstrate its robustness and scalability to complex real-world scenarios.
* **Fairness of evaluation strategy.** Two reviewers misunderstood our evaluation strategy as a comparison on 3D object-level segmentation, which may be unfair for those pixel-based baselines. We apologize for the confusion, the evaluation is actually done on 2D pixel-level, which aligns with the original setting of all the baseline methods, thereby ensuring a fair comparison.

During the discussion with Reviewer Yvnk, a concern was raised regarding the **real-world applicability** of our method, specifically, whether the single-object retrieval performance would degrade when applying the thresholding strategy. To address this, we conducted an additional experiment, which shows that our method maintains state-of-the-art performance even with thresholding applied.

We sincerely thank the reviewers once again for their time and valuable feedback.

---

### Decision · Program_Chairs · 2025-09-17

**Decision:**

Accept (poster)

**Comment:**

The paper introduces a 3D segmentation method that splits Gaussians used in 3D Gaussian splat scene representations into distinct object sets before reconstruction. Once the reconstruction is complete, the scene is naturally segmented into individual objects, achieving 3D segmentation. As a result, the method maintains precise object-Gaussian correspondence for improved 3D geometry and segmentation accuracy.

The paper received five reviews: four borderline accepts and one borderline reject. Positive reviewers highlighted several strengths: (a) the intellectual novelty of the “Segment-then-Splat” idea, (b) strong experimental results, and (c) natural compatibility with dynamic scenes. Weaknesses raised during the reviews and author--reviewer discussion included: (a) the technical contribution is somewhat limited and primarily related to class-agnostic segmentation, (b) increased preprocessing complexity, and (c) issues with robustness to large occlusions. The authors acknowledged these limitations.

Overall, four out of five reviewers felt that the significance of the “Segment-then-Splat” idea and the strong results outweighed the weaknesses. The AC concurs with this assessment and recommends acceptance. The authors are strongly encouraged to incorporate the additional experiments, results, and clarifications from the rebuttal and discussion into the final version. They should also carefully address cG1h’s remarks and the other weaknesses raised by the reviewers, particularly in the limitations section.